# Finding Wasserstein Ball Center: Efficient Algorithm and The Applications in Fairness

**Yuntao Wang** [* 1]  **Yuxuan li** [* 2]  **Qingyuan Yang** [1]  **Hu Ding** [1]

## Abstract

Wasserstein Barycenter (WB) is a fundamental geometric optimization problem in machine learning, whose objective is to find a representative probability measure that minimizes the sum of Wasserstein distances to given distributions. WB has a number of applications in various areas. However, WB may lead to unfair outcome towards underrepresented groups in some applications (e.g., a "minority" distribution may be far away from the obtained WB under Wasserstein distance). To address this issue, we propose an alternative objective called "Wasserstein Ball Center (WBC)". Specifically, WBC is a distribution that encompasses all input distributions within the minimum Wasserstein distance, which can be formulated as a "minmax" optimization problem. We show that the WBC problem with fixed support is equivalent to solving a large-scale linear programming (LP) instance, which is quite different from the previously studied LP model for WB. By incorporating some novel observations on the induced normal equation, we propose an efficient algorithm that accelerates the interior point method by $O(\min\{N^2m, Nm^2, m^4\})$ times ("$N$" is the number of distributions and "$m$" is the support size). Finally, we conduct a set of experiments on both synthetic and real-world datasets, demonstrating the computational efficiency of our algorithm, and showing its ability to provide more fairness for input distributions.

## 1. Introduction

To find a representative of several given probability distributions is a natural problem in machine learning. One popular approach is to compute the geometric center on probability space with induced distances between probabilities, such as the seminal *Wasserstein distance* in the field of optimal transport (Villani, 2021). Given a non-negative weight vector $(\omega_1, \omega_2, \ldots, \omega_N)$ with $N \geq 2$, the **Wasserstein barycenter** (WB) of $N$ probability measures $\{\mu_k\}_{k=1}^N$ under norm $p$ is defined as the weighted Fréchet Mean mean under Wasserstein distance. Namely, it is the solution of the following problem

$$\min_{\mu \in \mathcal{P}_p(\Omega)} \sum_{t=1}^N \omega_t W_p^p(\mu, \mu_t), \quad (1)$$

where $\mathcal{P}_p(\Omega)$ is the set of Borel probability measure on $\Omega$ with finite p-th moment, and $W_p$ is the Wasserstein distance under norm $p$, which will be formally defined in Section 2. WB has found various applications in many fields, such as economics (Carlier & Ekeland, 2010; Chiappori, 2017), physics (Benamou et al., 2014; Koehl et al., 2019), statistics (Kroshnin et al., 2021; Backhoff-Veraguas et al., 2022; Goldfeld et al., 2024), medical imaging (Gramfort et al., 2015; Nadeem et al., 2020), and machine learning (Qin et al., 2021; Cheng et al., 2021; Zhuang et al., 2022).

As the Fréchet mean under Wasserstein distance, WB tends to assign more measure to the region where the input density functions "cluster". In other words, to minimize the average distance from the barycenter to the input probabilities, if the support of most distribution is concentrated with high probability in a region, then the WB should also have measure concentrated in that region. But this property may behave "unfairly" to "minority", *i.e.,* the distributions with support deviated from the majority of others could be too far away from the WB. Fig. 1 gives an intuitive demonstration for this issue.

The unfairness could cause negative impact in some scenarios (Mehrabi et al., 2021; Caton & Haas, 2024). In medical applications, datasets usually reflect societal biases or historical inequities in healthcare access. If a deep learning model is trained on biased data, it can propagate or even amplify these biases, leading to unequal treatment outcomes. For

---
*Equal contribution [1]School of Computer Science and Technology, University of Science and Technology of China, Hefei, China [2]School of Artificial Intelligence and Data Science, University of Science and Technology of China, Hefei, China. Correspondence to: Hu Ding <huding@ustc.edu.cn>.

*Proceedings of the $42^{nd}$ International Conference on Machine Learning*, Vancouver, Canada. PMLR 267, 2025. Copyright 2025 by the author(s).

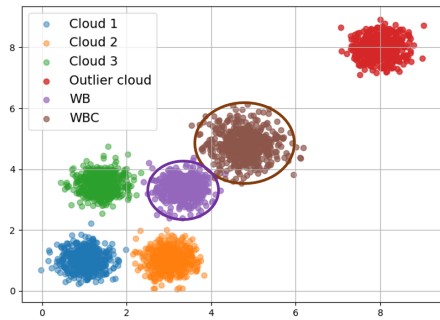 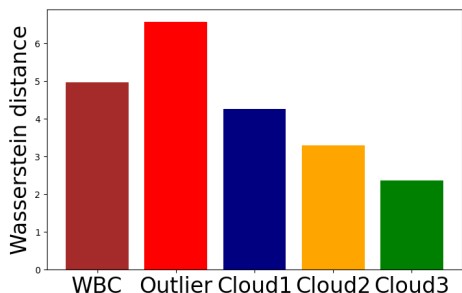

*Figure 1.* Left: four probability measures (cloud 1-3, and an outlier cloud), with their WB (computed by (1)) enclosed in purple ellipse, WBC (computed by (2)) enclosed in brown ellipse. The red cloud (*i.e.*, the outlier cloud) has measure distributed distinctly from the other three clouds. In the histogram on the right, the "WBC" bin denotes the maximal Wasserstein distance from WBC to four clouds, and each of the other four bins denotes the Wasserstein distance from WB to the corresponding cloud. From this histogram, we can see that WBC treats the outlier more equally (for example, the "WBC" bin is significantly lower than the "Outlier" bin).

instance, many skin lesion datasets predominantly contain images from lighter-skinned individuals (Bhardwaj & Rege, 2021; Adegun & Viriri, 2021). As a result, AI systems trained on such data often perform poorly on darker-skinned patients, leading to underdiagnosis or misdiagnosis in certain populations (Montoya et al., 2025). In addition, the medical data may come from different sources, therefore a fair alignment can also enhance model's accuracy (Aayushman et al., 2024; Lin et al., 2023).

To address this unfair issue, we propose a different objective function. Rather than minimizing the summation of Wasserstein distances, we try to find a distribution that is of minimal distance from the farthest input distribution:

$$\min_{\mu \in \mathcal{P}_p(\Omega)} \max_{1 \leq t \leq N} W_p(\mu, \mu_t). \tag{2}$$

From a geometric perspective, we can think of it as the center of the ball in Wasserstein space, who covers all input distributions with minimum radius. In this setting, the output distribution does not put extra measure to the region where input distributions cluster with high density. Please see Fig. 1 for an illustrative comparison. We call the solution for Problem (2) the **Wasserstein Ball Center (WBC)**, and aim to design an efficient algorithm to solve it. It should be noted that "Wasserstein ball" is not a new concept and actually has been studied by several works before (Yue et al., 2022; Pesenti & Jaimungal, 2023; Chen et al., 2024), yet these previous works usually assume **the ball center is given** and take the ball as a feasible region for constraining some optimization objective. In this paper, we focus on how to compute an optimal center so that the induced radius (under Wasserstein distance) is minimized.

The WBC model (2) also shares the similar "**minmax**" property with the recently proposed clustering problem under the notion of group fairness, called *social fairness* or *equitable group representation* (Abbasi et al., 2021; Ghadiri

et al., 2021). In this framework, the data set for clustering consists of a set of groups, where each group contains a set of data items. Standard clustering algorithms could incur higher clustering costs for certain protected groups (*e.g.,* the groups that are defined by a sensitive attribute such as gender or race). This lack of fairness has motivated the study of clustering that minimizes the maximum clustering cost across different demographic groups, such as gender, race, high socioeconomic status, *etc*. The sensitive groups thus can obtain more equitable attention in the algorithm. Comparing with the traditional clustering models (*e.g., $k$*-means or $k$-median clustering s), the main difference (which is also the main challenge) lies in this "minmax" objective.

## 1.1. Our Main Contributions

Similar with the aforementioned socially fair clustering problems, solving Problem (2) is also not easy due to its inherent "minmax" nature. When all distributions are of discrete support, the problem can be formulated as a linear programming (LP) problem, where the details are shown in Section 2. Partly inspired by the recent developments of *interior point method (IPM)* and their various applications (Gondzio, 2012) (*e.g.*, the IPM algorithm for solving WB (Ge et al., 2019)), we also consider developing an efficient IPM based algorithm for the WBC problem, though its formulation is much more complicated than WB due to the minimax issue.

Technically, there is a major challenge for directly applying IPM to the WBC problem, *viz.,* the computational cost and space complexity are both very large. The linear programming formulation of WBC has $m \sum_{i=1}^{N} m_i + m + N + 1$ variables and $Nm + \sum_{i=1}^{N} m_i + N + 1$ constraints, where the integer $N$ denotes number of distributions, $m_i$ and $m$ denote the support sizes for the $i$-th distribution and WBC respectively. This brings the challenge for computing the in-

ner loop of IPM, which requires a time complexity as large as $O((Nm + \sum_{i=1}^{N} m_i + N)^2 (m \sum_{i=1}^{N} m_i + m + N))$. To tackle this difficulty, we grind the intrinsic information of constraint matrix to simplify the Newton normal equation, which is a linear system with a large positive definite constraint matrix, and is also the most expensive part in each inner loop of IPM. Specifically, we simplify the matrix inverse occurred in the solution of *Newton path*, based on an important observation:

*After performing some carefully designed transformations, the seemingly dense coefficient matrix of normal equation (a core part in our IPM framework) can be converted into a block-diagonal matrix in a particularly efficient way, where the inverse of each block is also easy to obtain.*

Through this observation, we obtain an $\mathbf{O(Nm^3 + N^2m^2 + N^3)}$ time complexity for each iteration, whereas the vanilla IPM requires $\mathbf{O(N^3m^4)}$ by straight matrix inversion (for simplicity we just assume $m_i = O(m)$ here). The latter one is often beyond acceptable scope in real-world scenarios. For example, assume we are given an instance that $N = 1000$ and $m = 100$. In each loop of IPM, the complexity of our algorithm is $O(10^{10})$, while the vanilla IPM requires $O(10^{17})$, which is $O(10^7)$ times higher. The strict description on this result is presented in Theorem 3.2. As for the practical effectiveness, our algorithm is significantly faster than the popular commercial solver "Gurobi" (Gurobi Optimization, LLC, 2024), which provides a concurrent optimizer that run different state-of-the-art optimization methods simultaneously. In particular for LP models, Gurobi simultaneously runs the "dual simplex", "parallel barrier", and "primal simplex" algorithms in independent threads. In other words, our proposed algorithm runs faster than all these commonly used LP algorithms. In Section 4, we provide a set of experiments on numerical performance and applications.

### 1.2. Related works

We present an overview on several important related works, and the detailed introduction is placed in Section A.

**Wasserstein distance** Wasserstein distance is a classic topic in machine learning (Rüschendorf, 1985). Cuturi (2013) introduced the "Sinkhorn Distance", which incorporates an entropic regularization term to smooth the transportation problem, offering paralleled and significantly faster solutions than exact computation of the discrete Wasserstein distance. Following Cuturi's work, recent years have seen the development of several improved Sinkhorn algorithms (Lin et al., 2019; Altschuler et al., 2019; Benamou et al., 2015; Altschuler et al., 2017). The discrete Wasserstein distance is also closely related to the *min-cost max flow*

problem (Ahuja et al., 1994). In the past years, a number of elegant algorithms have been proposed in theoretical computer science, such as (Orlin, 1988; Lee & Sidford, 2014; Sherman, 2017), and the ones focusing on Euclidean space (Khesin et al., 2021; Fox & Lu, 2023) or parallelism (Lahn et al., 2023).

**Wasserstein barycenter.** The computation for Wasserstein barycenter is an NP-hard problem (Altschuler & Boix-Adsera, 2022). Cuturi & Doucet (2014) showed that the computation for WB can be improved by using an entropic regularization. Further progress includes iterative Bregman projection (IBP) algorithm (Benamou et al., 2015), the semi-dual gradient descent (Cuturi & Peyré, 2018), accelerated primal-dual gradient descent (APDAGD) (Kroshnin et al., 2019), alternating direction method of multipliers (ADMM) (Ye et al., 2017), deterministic IBP (Lin et al., 2020), and the IPM algorithm MAAIPM (Ge et al., 2019). In Euclidean space, WB can be solved in polynomial time with fixed dimension (Altschuler & Boix-Adsera, 2021; Agarwal et al., 2025).

**Interior Point Method.** Dikin (1967) proposed the first interior point method. Further, Karmarkar (1984) proposed the first polynomial time method for linear programming called "Karmarkar's algorithm". Mehrotra's *predictor-corrector* algorithm (Mehrotra, 1992) provides the basis for most implementations of this class of methods, which is also the type of IPM applied in this paper (the details of predictor-corrector IPM are presented in Section 3.2). The seminal Mizuno-Todd-Ye predictor-corrector method has quadratic convergence (Mizuno et al., 1993; Ye et al., 1993). For more details on IPM, we refer the reader to the survey paper (Gondzio, 2012). Recently, there are also several new studies on reducing the IPM complexity (Jiang et al., 2021; Cohen et al., 2021), which rely on a technique called "matrix maintenance" to reduce the update time of each iteration.

## 2. Preliminaries

**Some notations.** Throughout this paper, we always use low case letter to denote scalar value, and bold low case letter to denote vector; also, we use "$[n]$" to denote the set $\{1, 2, \cdots, n\}$. Let "$\mathbf{1}_m$" be the all-one vector in $\mathbb{R}^m$. For two discrete probability vectors $\boldsymbol{u} \in \mathbb{R}^{m_1}$ and $\boldsymbol{v} \in \mathbb{R}^{m_2}$ with $m_1$ and $m_2$ being their support sizes, define the set of matrices $\mathcal{M}(\boldsymbol{u}, \boldsymbol{v}) = \{\Pi \in \mathbb{R}_+^{m_1 \times m_2} : \Pi \mathbf{1}_{m_2} = \boldsymbol{u}, \Pi^\top \mathbf{1}_{m_1} = \boldsymbol{v}\}$ as the **coupling matrices**, which consists of all joint distributions of margin $\boldsymbol{u}$ and $\boldsymbol{v}$. Let $\mathcal{Q} = \{(a_i, \boldsymbol{q}_i) : i \in [m_1]\}$ denote the discrete probability measure supported on $m_1$ points $\boldsymbol{q}_1, \ldots, \boldsymbol{q}_{m_1}$ in $\mathbb{R}^d$ with weights $a_1, \ldots, a_{m_1}$ respectively. Another probability measure $\mathcal{P} = \{(b_j, \boldsymbol{p}_j) : j \in [m_2]\}$ is defined similarly. The **$p$-norm Wasserstein distance** of these two discrete proba-

bility measures $\mathcal{Q}$ and $\mathcal{P}$ is

$$W_p(\mathcal{Q},\mathcal{P}) := \min \big\{ \big( \sum_{i=1}^{m_1} \sum_{j=1}^{m_2} \pi_{ij} \|\boldsymbol{q}_i - \boldsymbol{p}_j\|_p^p \big)^{\frac{1}{p}} :$$
$$\Pi = (\pi_{ij})_{(m_1 \times m_2)} \in \mathcal{M}(\boldsymbol{a},\boldsymbol{b}) \big\} \quad \text{(OT)}$$

where $\boldsymbol{a} = (a_1,\ldots,a_{m_1})^\top$ and $\boldsymbol{b} = (b_1,\ldots,b_{m_2})^\top$. The coupling matrix achieving the minimum value can be seen as an alignment between the two distributions, which provides the optimal transportation (OT) plan between their supports.

Now, we consider the scenario where there are multiple probability measures in the same space, $\{\mathcal{P}^{(t)}, t = 1, \in [N]\}$ ($N \geq 2$), where each $\mathcal{P}^{(t)} = \{(a_i^{(t)}, \boldsymbol{q}_i^{(t)}) : i \in [m_t]\}$ with probability vector $\boldsymbol{a}^{(t)} = (a_1^{(t)}, \ldots, a_{m_t}^{(t)})^\top$. The optimal **Wasserstein ball center** (WBC) $\mathcal{P}_{\text{opt}} = \{(w_i, \boldsymbol{x}_i) : i \in [m]\}$ is another probability measure such that the maximum Wasserstein distance to these given $N$ probability measures is minimized, as defined in the objective function (2) when $\Omega = \{\boldsymbol{x}_1, \cdots, \boldsymbol{x}_m\}$. The probability $\boldsymbol{w} = (w_1, \cdots, w_m)^\top$ of $\mathcal{P}_{opt}$ and its coupling matrices with $\{\boldsymbol{a}^{(t)} : t \in [N]\}$ must be in the solution set $\mathcal{S} =$

$$\big\{ (\boldsymbol{w}, \Pi^{(1)}, \ldots, \Pi^{(N)}) \in \mathbb{R}_+^m \times \mathbb{R}_+^{m \times m_1} \times \cdots \times \mathbb{R}_+^{m \times m_N} :$$
$$\underbrace{\mathbf{1}_m^\top \boldsymbol{w} = 1, \boldsymbol{w} \geq 0}_{\text{constraint (i)}};$$
$$\underbrace{\Pi^{(t)} \mathbf{1}_{m_t} = \boldsymbol{w}, \big( \Pi^{(t)} \big)^\top \mathbf{1}_m = \boldsymbol{a}^{(t)}, \Pi^{(t)} \geq 0, t \in [N]}_{\text{constraint (ii)}} \big\}, \quad (3)$$

where "$\geq$" means each entry of the vector/matrix is non-negative, the constraint (i) guarantees $\boldsymbol{w}$ to be a feasible probability distribution, and the constraint (ii) follows the coupling condition in (OT). For a given support $\Omega$, the **distance matrices** are defined as $D^{(t)}(\Omega) = (\|\boldsymbol{x}_i - \boldsymbol{q}_j^{(t)}\|_p^p)_{(i,j)} \in \mathbb{R}^{m \times m_t}$ for $t = 1, \ldots, N$. Since in most practical applications, the measures in $\{\mathcal{P}^{(t)}\}_{t=1}^N$ have finite support points, and we can assume the barycenter is supported on a given finite set $\Omega$ (in general, $\Omega$ does not need to be the same as the supports of $\mathcal{P}^{(t)}$s), the problem WBC in (2) can be re-written as:

$$\min_{\boldsymbol{w}} \max_{t \in [N]} \min_{\Pi^{(t)}} \big\langle D^{(t)}, \Pi^{(t)} \big\rangle \quad \text{s.t. } (\boldsymbol{w}, \Pi^{(1)}, \ldots, \Pi^{(N)}) \in \mathcal{S}$$
$$(4)$$

where "$\langle \cdot, \cdot \rangle$" denotes the inner product in Euclidean space, and $D^{(t)}$ denotes $D^t(\Omega)$ for simplicity.

# 3. Our Optimization Framework for WBC

In this section, we introduce our optimization framework for solving the Problem (4). Specifically, in Section 3.1 we formalize Problem (4) to be the standard LP form, ensuring that the constraint matrix is full row-rank so that the

IPM framework introduced in Section 3.2 can run properly. In Section 3.3 we illustrate how to eliminate unnecessary computations to speed up IPM, as the key part of our contribution in the current article. We take advantage of some critical observation on the symmetry behind the seemingly dense constraint matrix. With that observation, we can significantly reduce the complexity of the most time consuming part in each loop of IPM, which is the inversing for a large matrix.

### 3.1. From Minmax to LP

First, we apply a slack variable $\gamma \in \mathbb{R}$, turning Problem (4) into the following LP problem:

$$\min_{\boldsymbol{w}, \Pi^{(t)}, \gamma} \gamma$$
$$\text{s.t. } (\boldsymbol{w}, \Pi^{(1)}, \ldots, \Pi^{(N)}) \in \mathcal{S}, \ \boldsymbol{x}_1, \ldots, \boldsymbol{x}_m \in \mathbb{R}^n \quad (5)$$
$$\big\langle D^{(t)}(X), \Pi^{(t)} \big\rangle \leq \gamma, \ 1 \leq t \leq N.$$

In (5), the value $\gamma$ indicates the maximal Wasserstein distance from the WBC to the given $N$ probability measures.

Then, we conduct the following transformation on Problem (5). We use "$\text{vec}(U)$" to denote the vectorization of a matrix $U$, which is a vector obtained by concatenating every column of $U$ to its predecessor. Let $\text{diag}(U_1, \cdots, U_m)$ denote the block diagonal matrix with the $i$-th block $U_i$, and "$\otimes$" denote tensor product. Let "$I_m$" be the identity matrix of order $m$. To transform the problem to a standard LP form, we vectorize the constraint (ii) in (3), "$\Pi^{(t)} \mathbf{1}_{m_t} = \boldsymbol{w}$" and "$(\Pi^{(t)})^\top \mathbf{1}_m = \boldsymbol{a}^{(t)}$", to be:

$$(\mathbf{1}_{m_t}^\top \otimes I_m) \text{vec}(\Pi^{(t)}) = \boldsymbol{w}, \ (I_{m_t} \otimes \mathbf{1}_m^\top) \text{vec}(\Pi^{(t)}) = \boldsymbol{a}^{(t)},$$

for $t = 1, \cdots, N$. We further define the following matrices: $E_1 = \text{diag}(I_{m_1} \otimes \mathbf{1}_m^\top, ..., I_{m_N} \otimes \mathbf{1}_m^\top)$, $E_2 = \text{diag}(\mathbf{1}_{m_1}^\top \otimes I_m, ..., \mathbf{1}_{m_N}^\top \otimes I_m)$, $E_3 = -\mathbf{1}_N \otimes I_m$ and $D = \text{diag}(\text{vec}(D^{(1)}), \ldots, \text{vec}(D^{(N)}))$. Through some simple calculations, Problem (5) can be formulated as a standard LP model:

$$\min \boldsymbol{c}^\top \boldsymbol{x} \quad \text{s.t. } A\boldsymbol{x} = \boldsymbol{b}, \boldsymbol{x} \geq 0 \quad \text{(WBC-LP)}$$

where $\boldsymbol{x} = (\text{vec}(\Pi^{(1)}); ...; \text{vec}(\Pi^{(N)}); \boldsymbol{w}; \gamma_1; \ldots; \gamma_N, \gamma)$, $\gamma_i = \gamma - \langle D^{(i)}, \Pi^{(i)} \rangle$, $\boldsymbol{b} = (\boldsymbol{a}^{(1)}; ... \boldsymbol{a}^{(N)}; \underbrace{\mathbf{0}_m; ...; \mathbf{0}_m}_{N}; 1)$,

$\boldsymbol{c} = (0; 0; \ldots; 0; 1)$ and

$$A = \begin{bmatrix} E_1 & & & \\ E_2 & E_3 & & \\ & \mathbf{1}_m^\top & & \\ D & & I_N & -\mathbf{1}_N \end{bmatrix}.$$

Let $M := \sum_{i=1}^N m_i$, and then the numbers of constraints and variables are $n_{\text{c}} := Nm + M + N + 1$ and $n_{\text{v}} :=$

$Mm + m + N + 1$, respectively. For a visual demonstration of these matrices, please see Eq. (12).

**Preprocessing on (WBC-LP).** To implement IPM, it is essential that the constraint matrix $A$ should be of full row-rank, so that the problem's complexity is reduced and IPM is also guaranteed to work more efficiently. In particular, the "full row-rank" property can simplify the procedure for solving the "normal equation" in IPM (which is discussed in next section). The following lemma eliminates all redundant constraints, turning $A$ into a full row-rank matrix $\bar{A}$. Specifically, $\bar{A} \in \mathbb{R}^{(n_c - N) \times n_v}$ is the matrix obtained from $A$ by removing the $(M+1)$-th, $(M+m+1)$-th, $\cdots$, $(M + (N-1)m + 1)$-th rows of $A$, and $\bar{b} \in \mathbb{R}^{n_c - N}$ is the vector obtained from $b$ by removing the $(M+1)$-th, $(M+m+1)$-th, $\cdots$, $(M + (N-1)m + 1)$-th entries of $b$.

**Proposition 3.1.** *1) $\bar{A}$ has full row-rank; 2) solving the equation $Ax = b$ in (WBC-LP) is equivalent to solving the equation $\bar{A}x = \bar{b}$.*

We leave the proof to Section B. Due to Theorem 3.1, we will focus on $\bar{A}$ instead of $A$ in the following sections.

### 3.2. Predictor-Corrector IPM on WBC-LP

We choose the classic predictor-corrector scheme (Mehrotra, 1992), which is a variant of primal-dual IPM, that applies the block coordinate descent on the Lagrangian of the linear program smoothed by logarithm regularization. As a second order method, it is proved to have quadratic convergence rate (Ye et al., 1993), which surpasses first-order methods in general (for a detailed analysis, please see (Wright, 1997)).

**The high level idea.** When we deal with a primal-dual system of linear programming, from the *Karush–Kuhn–Tucker* theory (Wright, 1997), we can update the primal and dual variables step by step using a Newton-like method to the Lagrangian of LP. In particular, the algorithm takes a logarithm barrier function, which is determined by a coefficient $\mu_+ \geq 0$, to measure how close the current solution is to that of the original problem described in WBC-LP.

Let $y \in \mathbb{R}^{n_c - N}$ and $s \in \mathbb{R}^{n_v}$ be the dual variables corresponding to the constraints "$\bar{A}x = \bar{b}$" and "$x \geq 0$", respectively. Writing in matrix form, the search direction at the current primal-dual tuple $(x, y, s)$ should be the solution of the following nonlinear system of equations:

$$\begin{bmatrix} 0 & \bar{A}^\top & I_{n_v} \\ \bar{A} & 0 & 0 \\ S & 0 & X \end{bmatrix} \begin{bmatrix} \Delta x \\ \Delta y \\ \Delta s \end{bmatrix} = - \begin{bmatrix} \bar{A}^\top y + s - c \\ \bar{A}x - b \\ Xs + \Delta X \Delta s + \mu_+ \mathbf{1}_{n_v} \end{bmatrix} \quad (6)$$

where the notation "$\Delta$" denotes the change when updating some variable or matrix. For the vector $x$, in Eq. (6) we use the capitalized letter "$X$" to denote the diagonal matrix with $X_{ii} = x_i$, where $x_i$ is the $i$-th term of $x$. Similarly, we also have the diagonal matrix $S$ for the vector $s$ in (6).

Because of the quadratic term "$\Delta X \Delta s$" on the right hand-side (RHS), Eq. (6) does not have an explicit solution. To reduce this nonlinear system to be a linear case, first we obtain a **predictor step** by removing the "$\Delta X \Delta s + \mu_+ \mathbf{1}_{n_v}$" term on the RHS of Eq. (6), then compute the **corrector step** by assigning the solution of predictor step to the quadratic term in the RHS of Eq. (6). Through combining them together, we obtain the final update step, hence the name "predictor-corrector". The value $\mu_+$ is updated after each loop, descending as the primal-dual gap narrows. When $\mu_+$ approaches to 0, the current position convergences to the optimal solution of WBC-LP.

Built upon the above idea, both the predictor and corrector steps can be obtained by sequentially computing $\Delta y, \Delta s$ and $\Delta x$, where the complete details are shown in Algorithm 2 of Appendix C. Here, we only focus on the most time consuming part, computing $\Delta y$, as the solution of **normal equation** (Wright, 1997). Namely, for both the predictor and corrector steps, we need to solve a system of linear equations as the following form:

$$(\bar{A}R\bar{A}^\top)\Delta y = f, \quad (7)$$

where $R = \texttt{diag}(s)^{-1}X$, and the symbol "$f$" represents the vector obtained in either the predictor or corrector steps, as shown in Eqs. (13) and (14) of Algorithm 2 (here, we can temporarily ignore the detailed form of $f$ since it does not affect designing our algorithm for solving Eq. (7)).

Other parts in one loop of Algorithm 2 are just simple operations like matrix-vector multiplications, and thus the whole time complexity is dominated by the cost for solving Eq. (7). We can solve Eq. (7) by inverting the matrix applied to $\Delta y$ (this requires $\bar{A}$ to be of full row-rank, and this is also the reason why we need Theorem 3.1). However, ordinary inverse to such a large matrix will take excessively high complexity, as shown in Remark 3.3 later. This motivates us to propose a new method to further improve the complexity in next section.

### 3.3. Solving the Normal Equation (7) Efficiently

We introduce an efficient algorithm for solving the normal equation $(\bar{A}R\bar{A}^\top)\Delta y = f$ of Eq. (7), whose complexity is summarized in the following theorem.

**Theorem 3.2.** *In each inner iteration of IPM, the time complexity in terms of flops is $O(m^2 \sum_{i=1}^{N} m_i + Nm^3 + N^2m^2 + N^3)$, and the memory usage in terms of doubles is $O(m \sum_{i=1}^{N} m_i + Nm^2 + N^2)$.*

*Remark* 3.3. For comparison, a vanilla IPM takes $O((Nm + \sum_{i=1}^{N} m_i + N)^2(m \sum_{i=1}^{N} m_i + m + N))$ time and $O((Nm + \sum_{i=1}^{N} m_i + N)(m \sum_{i=1}^{N} m_i + m + N))$ storage.

**Roadmap of the proof for Theorem 3.2.** It is worth noting that we do not need to really compute the matrix $(\bar{A}R\bar{A}^\top)^{-1}$

for solving Eq. (7); instead, we only need to find an efficient way to simplify the structure of $\bar{A}R\bar{A}^\top$, so that we can easily compute the matrix-vector multiplication "$(\bar{A}R\bar{A}^\top)^{-1}\boldsymbol{f}$". Proposition 3.4 illustrates the detailed structure of $\bar{A}R\bar{A}^\top$, which paves the way for designing the following transformations. Our method contains three steps. First, we transform $\bar{A}R\bar{A}^\top$ to be a nearly block-diagonal matrix. Then, we resolve the most challenging part that hinders it from being block-diagonal, as shown in Theorem 3.6. Finally, we obtain the desired block-diagonal form for $\bar{A}R\bar{A}^\top$. Namely, to compute the product $(\bar{A}R\bar{A}^\top)^{-1}\boldsymbol{f}$, we only need to care about the inverses of several small matrices with sizes no bigger than $\max\{m_i, m, N\}$. As a consequence, we save the massive time and storage for inverting a matrix of size $O(Nm + N\max_i(m_i))$.

Let $\boldsymbol{r}$ be the $n_v$-dimensional vector with its $i$-th entry $r_i = R_{ii}$ (recall that $n_v$ denotes the number of variables as defined in Section 3.1). Also recall that $M = \sum_{i=1}^N m_i$, and we further define $M' = N(m-1)$. Let $\boldsymbol{z} = \boldsymbol{r}(Mm+2 : Mm+m)$, the sub-vector of $\boldsymbol{r}$ including from the $(Mm+2)$-th to $(Mm+m)$-th entries.

**Proposition 3.4.** *$\bar{A}R\bar{A}^T$ has the following form:*

$$\bar{A}R\bar{A}^T = \begin{bmatrix} B_1 & B_2 & \mathbf{0} & K_1 \\ B_2^\top & B_3 + B_4 & \boldsymbol{\alpha} & K_2 \\ \mathbf{0} & \boldsymbol{\alpha}^\top & l & 0 \\ K_1^\top & K_2^\top & 0 & W \end{bmatrix}$$

*where the sub-matrices are defined as follows (their illustrations are shown in Section D):*

- $B_1 \in \mathbb{R}^{M \times M}$ *is a diagonal matrix with positive diagonal entries; $B_2 \in \mathbb{R}^{M \times M'}$ is a block-diagonal matrix with $N$ blocks, where each $i$-th block is of size $m_i \times (m-1)$; $B_3 \in \mathbb{R}^{M' \times M'}$ is a diagonal matrix with positive diagonal entries; $B_4 = (\mathbf{1}_N \mathbf{1}_N^\top) \otimes \mathtt{diag}(\boldsymbol{z})$;*

- $\boldsymbol{\alpha} = -\mathbf{1}_N \otimes \boldsymbol{z}$; $l = \mathbf{1}_m^\top \boldsymbol{r}(n_v - m + 1 : n_v - N)$;

- $K_1 \in \mathbb{R}^{M \times N}$ *is a block-diagonal matrix with $N$ blocks, with the $i$-th block being of size $m_i \times 1$; $K_2 \in \mathbb{R}^{M' \times N}$ is a block-diagonal matrix with $N$ blocks, with the $i$-th block being of size $(m-1) \times 1$;*

- $W = W_1 + r_{n_v} \mathbf{1}_N \mathbf{1}_N^\top$, *where $r_{n_v}$ is the $n_v$-th term of $\boldsymbol{r}$, and $W_1 \in \mathbb{R}^{N \times N}$ is a diagonal matrix with positive diagonal entries.*

We left the proof for Proposition 3.4 to Appendix D. Then, we illustrate our idea for efficiently solving the normal equation (7) via three steps.

**Step 1: transforming $\bar{A}R\bar{A}^\top$ to be nearly block-diagonal.** We simplify the coefficient matrix $\bar{A}R\bar{A}^\top$ by performing

several transformations. Relying on Proposition 3.4 we design the transformation matrix $Q := \begin{bmatrix} I_M & & & \\ & I_{M'} & & \\ & & 1 & \\ -K_1^\top B_1^{-1} & & & I_N \end{bmatrix}$.

$\begin{bmatrix} I_M & & & \\ & I_{M'} & -\boldsymbol{\alpha}/l & \\ & & 1 & \\ & & & I_N \end{bmatrix} \cdot \begin{bmatrix} I_M & & & \\ -B_2^\top B_1^{-1} & I_{M'} & & \\ & & 1 & \\ & & & I_N \end{bmatrix}$, such that

$$Q \cdot \bar{A}R\bar{A}^T \cdot Q^\top = \begin{bmatrix} B_1 & & & \\ & G_1 + G_2 & & \overline{K}_2 \\ & & l & \\ & \overline{K}_2^\top & & \overline{W} \end{bmatrix}, \quad (8)$$

where $G_1 := B_3 - B_2^\top B_1^{-1} B_2$, $G_2 := B_4 - \frac{1}{l}\boldsymbol{\alpha}\boldsymbol{\alpha}^\top$, $\overline{W} = W - K_1^\top B_1^\top K_1$. See Section E for computation.

**Step 2: dealing with $(G_1 + G_2)^{-1}$.** In the RHS of Eq. (8), we can see that the transformed matrix is still not block-diagonal due to the existence of "$\overline{K}_2$". To further eliminate $\overline{K}_2$, we need to design another transformation matrix that involves the computation of $(G_1 + G_2)^{-1}$, the inverse of a large matrix of size $M' \times M'$. To find a way to compute $(G_1 + G_2)^{-1}$ more efficiently, we have a useful observation on the matrices $G_1$ and $G_2$, which indicates the specificity of their structures (the proof of Proposition 3.5 is placed in Section F).

**Proposition 3.5.** *There exist positive definite matrices $Y, G_{ii} \in \mathbb{R}^{(m-1) \times (m-1)}, i = 1, \dots, N$, such that $G_1 = \mathtt{diag}(G_{11}, \cdots, G_{NN})$ and $G_2 = (\mathbf{1}_N \mathbf{1}_N^\top) \otimes Y$.*

With the above observation, we achieve the following key lemma.

**Lemma 3.6.** *Let $\bar{Z} = \mathtt{diag}(\boldsymbol{z}) - \frac{1}{l}\boldsymbol{z}\boldsymbol{z}^\top$ ($\boldsymbol{z}$ is defined before Theorem 3.4), and $\tilde{G} = (\mathbf{1}_N \mathbf{1}_N^\top) \otimes (\bar{Z}^{-1} + \sum_{i=1}^N G_{ii}^{-1})^{-1}$, then we have*

$$(G_1 + G_2)^{-1} = G_1^{-1} - G_1^{-1}\tilde{G}G_1^{-1}. \quad (9)$$

*Further, the flops needed for applying a vector to the RHS of Eq. (9) is $O(Nm^2)$.*

*Proof.* Since $Y$ is positive definite, there should exit a matrix $U \in \mathbb{R}^{(m-1) \times (m-1)}$, such that $Y = U^\top U$. Then, from Proposition 3.5 we have $G_2 = (\mathbf{1}_N \mathbf{1}_N^\top) \otimes Y = (\mathbf{1}_N \otimes U^\top)(\mathbf{1}_N^\top \otimes U)$, and it implies

$$(G_1 + G_2)^{-1} = \left(G_1 + (\mathbf{1}_N \otimes U^\top) \cdot (\mathbf{1}_N^\top \otimes U)\right)^{-1}$$
$$= G_1^{-1} - G_1^{-1} \cdot (\mathbf{1}_N \otimes U^\top) \cdot$$
$$\left(I_{m-1} + (\mathbf{1}_N^\top \otimes U)G_1^{-1}(\mathbf{1}_N \otimes U^\top)\right)^{-1} \cdot (\mathbf{1}_N^\top \otimes U) \cdot G_1^{-1},$$

where the second equality is obtained by *Woodbury matrix identity* (Hager, 1989), a commonly used equation in linear algebra (see Section H). Noticing that both $G_1$ and $\mathbf{1}_N^\top \otimes U$ are divided into $N \times N$ sub-matrices of size

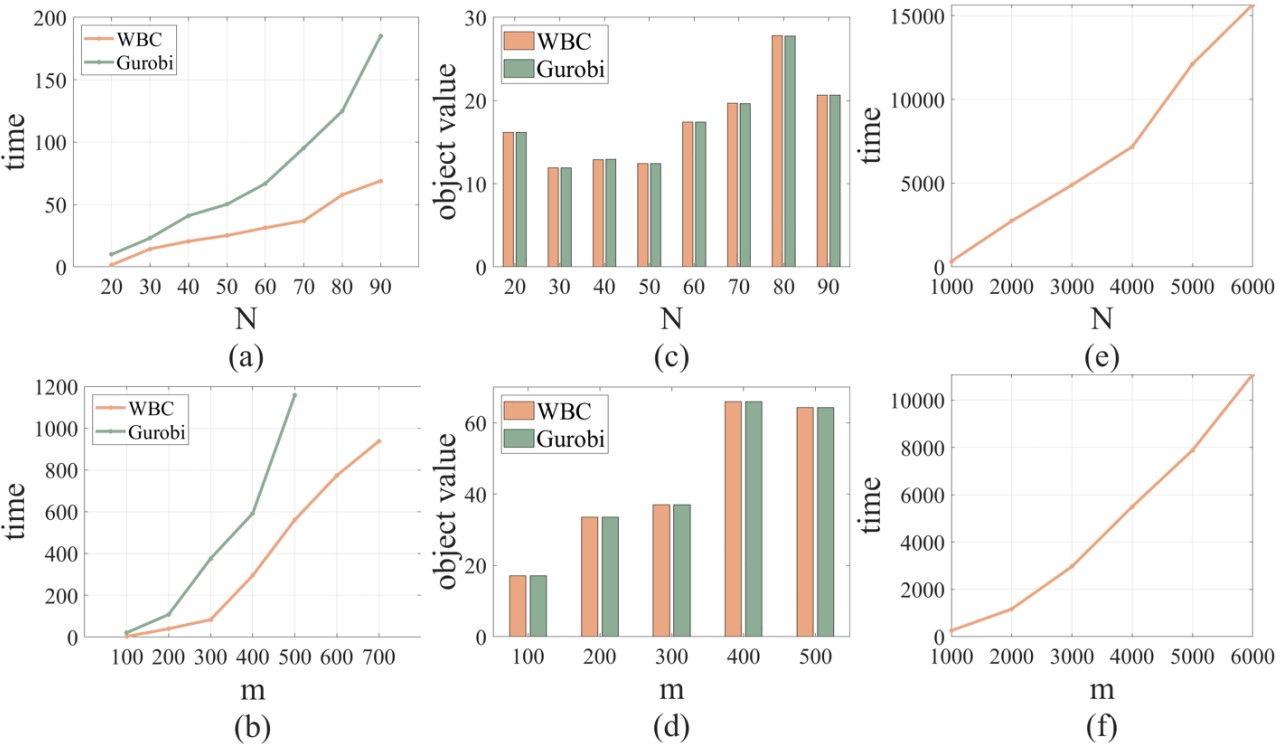

*Figure 2.* The first two column figures are the computational time (s) and obtained objective value of Gurobi and our method. For (a) and (c), we fix $m = 100$ and vary $N$. For (b) and (d), we fix $N = 30$ and vary $m$. The third column figures are the computational times of our method when the problem scale is very large. For (e), we fix $m = 50$. For (f), we fix $N = 10$.

$(m-1) \times (m-1)$, together with Proposition 3.5, we can simplify the RHS of the above equality to be

$$G_1^{-1} - G_1^{-1}\Big((\mathbf{1}_N \mathbf{1}_N^\top) \otimes$$

$$\big(U^\top (I_{m-1} + \sum_{i=1}^{N} UG_{ii}^{-1}U^\top)^{-1}U)\big)G_1^{-1}$$

$$=G_1^{-1} - G_1^{-1}\Big((\mathbf{1}_N \mathbf{1}_N^\top) \otimes (Y^{-1} + \sum_{i=1}^{N} G_{ii}^{-1})^{-1}\Big)G_1^{-1}.$$

Therefore, we obtain the equality (9).

As for the complexity, we notice the following two facts. **(1)** $G_1^{-1}$ is a block diagonal matrix with only $N(m-1)^2$ non-zero terms, thus the operation of multiplying a vector to it takes no more than $O(Nm^2)$ time. **(2)** The matrix $\tilde{G}$ can be viewed as an $N \times N$ matrix with every term being $(Z^{-1} + \sum_{i=1}^{N} G_{ii}^{-1})^{-1}$. That is, we only have to store $(Z^{-1} + \sum_{i=1}^{N} G_{ii}^{-1})^{-1}$, a square matrix of size $m-1$ instead of $\tilde{G}$ with size $N(m-1)$. Immediately, if we multiply a vector on the right of $\tilde{G}$, the $i$-th coordinate of the resulting vector would be the same as the $(i+m-1)$-th coordinate for any $i \leq M' - (m-1)$. Therefore we only have to compute

the first $m-1$ rows, then copy them $N$ times down to obtain the multiplication. This part also takes $O(Nm^2)$ time. □

**Step 3: from nearly block-diagonal to block-diagonal, and achieve the final solution $\Delta y$.** We design another transformation matrix $Q'$ to eliminate $\overline{K}_2$ in the nearly block diagonal matrix obtained in (8). Specifically, we let $Q' = \begin{bmatrix} I_M & & & \\ & I_{M'} & & \\ & & 1 & \\ & -\overline{K}_2^\top (G_1+G_2)^{-1} & & I_N \end{bmatrix}$. Through combining with the transformation $Q$ introduced in (8), we have

$$Q' \cdot Q\bar{A}R\bar{A}^T Q^\top \cdot Q'^\top = \mathtt{diag}(B_1, G_1 + G_2, l, \tilde{W}), \quad (10)$$

where $\tilde{W} = \overline{W} - \overline{K}_2^\top (G_1 + G_2)^{-1}\overline{K}_2$. Based on Lemma 3.6, the construction of $\tilde{W}$ can be done in $O(N^2 m^2)$ flops, since $\overline{K}_2$ has $N$ columns.

Now, we can solve the normal equation $(\bar{A}R\bar{A}^\top)\Delta y = f$ of Eq. (7). Set $\bar{y}$ as the solution to

$$\mathtt{diag}(B_1, G_1 + G_2, l, \tilde{W})\bar{y} = Q'Qf, \quad (11)$$

where the left coefficient matrix is from the obtained block-diagonal matrix in (10). Similar with the method for computing $\tilde{W}$, due to the term "$G_1 + G_2$" in the coefficient matrix

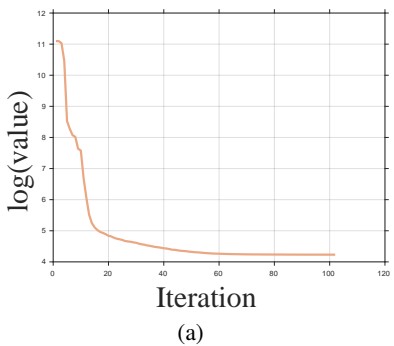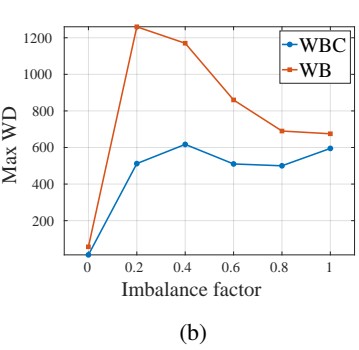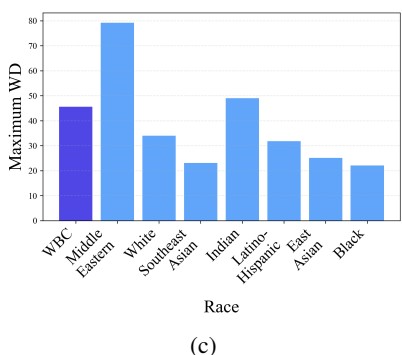

(a)            (b)            (c)

*Figure 3.* (a) $N = 90, m = 2000$. The objective value of our algorithm becomes stable after 80 iterations. For other settings of m and N, see Section J. (b) The maximum Wasserstein distance between the barycenter (WBC or WB) and input distributions, with fixing $N = 20, m = 500$. (c) The dark bin marked "WBC" is the maximum WD between WBC and all the faces, while the rest represent the maximum WD from WB to the faces in different races.

of $\bar{\boldsymbol{y}}$, we also need to apply Theorem 3.6 to obtain $\bar{\boldsymbol{y}}$. After that, from Eq. (10) it is easy to verify that $\Delta \boldsymbol{y} = Q^\top Q'^\top \bar{\boldsymbol{y}}$.

Algorithm 1 summarizes the above steps. Due to the space limit, we leave the detailed proof for the overall complexities to Section G.

---

**Algorithm 1** Solver for $(\bar{A}R\bar{A}^\top)\Delta \boldsymbol{y} = \boldsymbol{f}$

---

**Input:** $R \in \mathbb{R}^{n_v \times n_v}, \boldsymbol{f} \in \mathbb{R}^{n_c - N}$ as described in Eq. (7)
**Output:** The solution $\Delta \boldsymbol{y}$
Step 1: Compute blocks of $\bar{A}R\bar{A}^\top$
 1: Compute $B_1, B_2, B_3, K_1, K_2, W$
Step 2: Simplify the constraint matrix into block diagonal
 1: Compute $Q$
 2: $\overline{K}_2 \leftarrow K_2 - B_2^\top B_1^{-1} K_1$
 3: $\overline{W} \leftarrow W - K_1^\top B_1^{-1} K_1$        //Eliminate $K_1$
 4: Compute $G_1, G_2$
 5: Decompose $(G_1 + G_2)^{-1}$ according to Lemma 3.6
 6: Compute $Q'$
 7: $\tilde{W} \leftarrow \overline{W} - \overline{K}_2^\top (G_1 + G_2)^{-1} \overline{K}_2$    //Eliminate $\overline{K}_2$
Step 3: Solve $\Delta \boldsymbol{y}$
 1: $\boldsymbol{z}_1 \leftarrow Q\boldsymbol{f}, \boldsymbol{z}_2 \leftarrow Q'\boldsymbol{z}_1$
 2: $\boldsymbol{z}_3(1 : M) \leftarrow B_1^{-1}\boldsymbol{z}_2(1 : M)$
 3: $\boldsymbol{z}_3(n_c - N + 1 : n_c) \leftarrow \tilde{W}^{-1}\boldsymbol{z}_2(n_c - N + 1 : n_c)$
 4: $\boldsymbol{z}_3(n_c - N) \leftarrow c^{-1}\boldsymbol{z}_2(n_c - N)$
 5: Solve $(G_1 + G_2)\boldsymbol{z}_4(M+1 : n_c - N - 1) = \boldsymbol{z}_3(M+1 : n_c - N - 1)$
 6: $\boldsymbol{z}_5 \leftarrow Q'^\top \boldsymbol{z}_4, \Delta \boldsymbol{y} \leftarrow Q^\top \boldsymbol{z}_5$
**Return:** $\Delta \boldsymbol{y}$

---

## 4. Experiments

We conduct four experiments to investigate the real performance of our algorithm. **(1)** The first experiment demonstrates our advantages on computational speed and memory usage over the commercial solver Gurobi, a powerful optimization solver widely used across various fields such as operations research, finance, and data science. The baseline we choose is *Gurobi Optimizer version 11.0.0* (academic license). **(2)** The second experiment reflects the fairness of WBC over the standard WB. For these two experiments, we generate random datasets in Euclidean space, and the weights of $(q_1^{(t)}, ..., q_m^{(t)})$ in each distribution $\mathcal{P}^{(t)}$ are generated uniformly at random. **(3)** The third experiment further illustrates the performance on a real-world dataset FairFace (Karkkainen & Joo, 2021) with diverse racial representation. We choose 700 (100 for each race) images including seven racial groups of "Black", "East Asian", "Indian", "Latino-Hispanic", "Middle Eastern", "Southeast Asian" and "White" as 700 distributions, where each image actually can be regarded as a distribution of size $50 \times 50$ in $\mathbb{R}^2$. **(4)** The fourth experiment on 3D point-cloud gives a visual example to the fairness of WBC as a barycenter. All the experiments are implemented on a workstation, Intel(R) Core(TM) i5-9400 CPU @ 2.90GHz and 8GB for RAM, equipped with win64 - Windows 11+.0.

**(1) Comparison with Gurobi.** We set $m$ of all distributions to be equal for brevity. As shown in Fig. 2, our algorithm is always faster than Gurobi, and the gap between the two methods is expanding as the scale increases. Moreover, Gurobi cannot solve the instance with $m > 500$ due to memory limitation, which showcases the advantage of our space complexity in Theorem 3.2. We also illustrate the convergence speed of our algorithm. From Fig. 3(a) (and Figs. 5 to 8 in appendix), we can see that our algorithm converges in a super-linear rate, which is consistent with the theoretical result of (Ye et al., 1993).

**(2) Fairness of WBC.** To evaluate the performance on fairness, we first compare the maximum Wasserstein distance (Max WD) between the obtained centers and input distri-

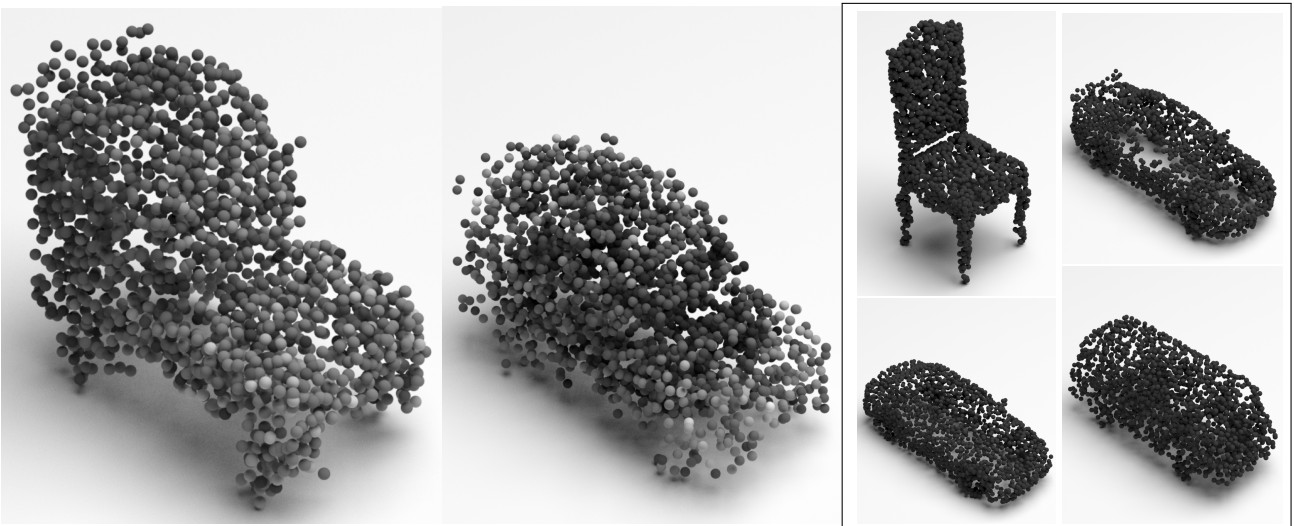

*Figure 4.* WBC (left) and WB (middle) of 3 cars and one chair. WB looks just like a car while our proposed WBC preserves characteristics of a chair, such as backrest and legs, indicating that WBC indeed places greater emphasis on fairness for the minority than WB.

butions. The experimental distributions are partitioned into two families, with each family's measure concentrated in distinct regions that are quite distant from each other. To quantify the disparity between these two families, we define the *imbalanced factor* ("imf") as the ratio of the smaller family's size to the larger one. For instance, "imf = 1" indicates that both families contain an equal number of distributions. As is shown in Fig. 3(b), the smaller the imf, the bigger the difference of the Max WD between WB and WBC (except for the case "imf=0", since the smaller family is empty in this case), indicating WBC's potential as the alternative to WB when fairness is concerned. Please see Section J.2 for the details of this part.

Another fairness measurement is the variance of Wasserstein distances from the obtained center to the input distributions: $\mathrm{var}(W(\bar{\mu}, \mu_i)) := \frac{1}{n-1} \sum_1^n (W(\bar{\mu}, \mu_i) - \overline{W})^2$, denoted as var(WD), where $\overline{W} = \frac{1}{n} \sum_{i=1}^N W(\bar{\mu}, \mu_i)$. Smaller var(WD) indicates reduced representative disparity among all inputs. Across all 2,700 experiments (detailed in Section J.2), WBC consistently achieves lower var(WD) compared to WB. For example, in $L_1$ space of dimension 100, we sample the 30 supports uniformly in a cube with side length 3, then translate 3 supports by adding $-\mathbf{1}_{100}$, and translate 5 supports by adding $2 * \mathbf{1}_{100}$, we have var(WD)=58.35 for WB , while var(WD)=418.61 for WBC. Those observations suggest that WBC not only improves fairness for the minority distributions, but also could enhance fairness across all distributions by limiting disparities between transport plans.

**(3) Experiments on FairFace Dataset.** From Fig. 3(c) , we can observe that the standard WB has a significant gap between the object values in different races. The object value of WB corresponding to "Middle Eastern" is 78.30,

which is far greater than the object value of our algorithm (45.37). Meanwhile, our algorithm achieves the object value only slightly higher than the average of WD to WB (38.70).

**(4) 3D Point-cloud Averaging.** WBC provides a means for computing the average shape of point-cloud data by casting a point cloud into an empirical probability measure. We take the ShapeNet Core-55 dataset (Chang et al., 2015), with each cloud containing 2048 points in $\mathbb{R}^3$. As an example, in Fig. 4 we show that compared with WB, the WBC preserves more characteristics of the "chair" (which can regarded as an outlier distribution from the other "car" clouds).

## 5. Conclusion

In this paper, we present an efficient algorithm to compute Wasserstein ball center, an alternative of Wasserstein barycenter with emphasis on minority distributions, which makes it more suitable for the tasks that are sensitive to fairness. For future work, it would be interesting to explore further properties and applications of WBC. Moreover, it is also deserved to study other models for fair ensembling different distributions in the field of machine learning.

## Acknowledgements

We want to thank the anonymous reviewers for their constructive comments. This work was supported in part by the National Key Research and Development Program of China (NO.2021YFA1000900), the National Natural Science Foundation of China (NO.62272432, NO.62432016) and the Natural Science Foundation of Anhui Province (NO.2208085MF163).

## Impact Statement

The proposed algorithm provides an effective alternative to the Wasserstein barycenter by offering a trade-off between transport efficiency and fairness. However, practitioners should be cautious of potential outliers arising from noise or data errors. To ensure robust performance, data cleansing or selection techniques should be applied prior to implementing the WBC.

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

## A. Related Works

**Wasserstein distance** The Wasserstein distance, also known as the Earth Mover's distance when $p = 1$, quantifies the disparity between two probability distributions, particularly when their supports are discrete sets. Computing the discrete Wasserstein metric reduces to solving a *minimum-cost maximum-flow* optimization problem (Ahuja et al., 1994; Orlin, 1988). Consider a bipartite graph with $n$ vertices, $m$ edges, and maximum edge weight $U$. Lee & Sidford (2014) designed an efficient general LP solver that can solve the minimum cost flow problem in $O(n^{2.5} poly(\log U))$. Lahn et al. (2023) designed a combinatorial algorithm that computes an $\epsilon$-approximate transport plan in $O(\frac{n^2}{\epsilon^2})$ sequential time and $O(\frac{\log n}{\epsilon^2})$ parallel time. Recently, Fox & Lu (2023) presented the first deterministic algorithm that computes optimal transportation in Euclidean space with near-linear time.

Wasserstein distance is also a classic topic in machine learning (Rüschendorf, 1985; Pele & Werman, 2009; Dognin et al., 2019). By using matrix scaling technique, Cuturi (2013) introduced the "Sinkhorn Distance", which incorporates an entropic regularization term to smooth the transportation problem, offering paralleled and significantly faster solutions than exact computation of the discrete Wasserstein distance. Following Cuturi's work, recent years have seen the development of several improved Sinkhorn algorithms (Lin et al., 2019; Altschuler et al., 2019; Benamou et al., 2015; Altschuler et al., 2017). Another practical variant is "sliced Wasserstein distance", which saves computation time by taking expectation of Wasserstein distance of the distributions projected from high dimension to one-dimensional space (Kolouri et al., 2019; Nietert et al., 2022).

**Wassertein barycenter.** Different from Wasserstein distance, the computation for Wasserstein barycenter is an NP-hard problem (Altschuler & Boix-Adsera, 2022). Cuturi & Doucet (2014) showed that the computation for WB can be improved by using an entropic regularization, leading to a simple gradient-descent scheme that was later improved and generalized under the iterative Bregman projection (IBP) algorithm (Benamou et al., 2015). Further progress includes the semi-dual gradient descent (Cuturi & Peyré, 2018), accelerated primal-dual gradient descent (APDAGD) (Kim et al., 2025; Kroshnin et al., 2019), alternating direction method of multipliers (ADMM) (Ye et al., 2017), deterministic IBP (Lin et al., 2020), and the IPM algorithm MAAIPM (Ge et al., 2019). In Euclidean space, WB can be solved in polynomial time with fixed dimension (Altschuler & Boix-Adsera, 2021). Vaskevicius & Chizat (2024) offers the first non-asymptotic convergence guarantees for approximating Wasserstein barycenter between discrete point clouds in the free-support/grid-free setting. Agarwal et al. (2025) gives the first polynomial algorithm for the case where the cost is $L_1$-norm. In applications, we sometimes need the support of WB to be sparse (Ye et al., 2017). Borgwardt (2022) proposed a strongly-polynomial 2-approximation algorithm for sparse Wasserstein barycenters. Yang & Ding (2024) investigated the case with outliers, connecting this problem with k-means clustering.

**Interior Point Method.** Dikin (1967) proposed the first interior point method, which was reinvented in Karmarkar (1984) which also proposed the first poly-time method for linear programming called "Karmarkar's algorithm". Since then IPM has attracted a great amount of attention, where one of the most successful IPM methods is the class of *primal-dual* approaches, which iteratively optimize the primal and the dual variable, until the dual gap converges to zero. In particular, Mehrotra's *predictor-corrector* algorithm (Mehrotra, 1992) provides the basis for most implementations of this class of methods, which is also the type of IPM applied in this paper (the details of predictor-corrector IPM are presented in Section 3.2). The seminal Mizuno-Todd-Ye predictor-corrector method has quadratic convergence (Mizuno et al., 1993; Ye et al., 1993). For more details on IPM, we refer the reader to the survey paper (Gondzio, 2012). Recently, there are also some new studies on reducing the exponent of IPM complexity in theoretical computer science (Jiang et al., 2021; Cohen et al., 2021), which relies on a technique called "matrix maintenance" to reduce the update time of each iteration.

**Fairness.** The fairness issue has attracted a great amount of attention in computer science (Joseph et al., 2016; Mehrabi et al., 2021; Caton & Haas, 2024). The proposed solutions include adjusting labels from sensitive groups to reconstruct unbiased mapping (Dwork et al., 2012; Jiang & Nachum, 2020), and removing sensitive attributions (Krasanakis et al., 2018). As is mentioned before, the WBC problem is also related with socially fair clustering (Abbasi et al., 2021; Ghadiri et al., 2021). An $(e^{O(p)} \frac{\log \ell}{\log \log \ell})$-approximation algorithm for socially fair clustering with the $\ell_p$-objective is given by Makarychev & Vakilian (2021). Unfairness can result from the issue of representation bias, which arises due to insufficient amount of data in certain groups or subgroups (Lohaus et al., 2020; Chai & Wang, 2022). Existing methods include fair data generation (Jang et al., 2021), multi-objective optimization (Martinez et al., 2020) and boosting (Gong & Kim, 2017). There are other concepts on fairness studied in computer science, for example the problems of fair k-means (Chierichetti et al., 2017; Chen et al., 2019; Backurs et al., 2019; Song et al., 2025) and fair regression (Agarwal et al., 2019; Wang et al., 2024).

# B. Proof of Theorem 3.1

We reformulate $A$ as the following,

$$
A = \begin{bmatrix}
F_1 & & & & & & & & \\
& F_2 & & & & & & & \\
& & \ddots & & & & & & \\
& & & F_N & & & & & \\
G_1^{(0)} & & & & & & -I_m & & \\
& G_2^{(0)} & & & & & -I_m & & \\
& & \ddots & & & & \vdots & & \\
& & & G_N^{(0)} & & & -I_m & & \\
& & & & & & \mathbf{1}_m^T & & \\
D_1 & & & & & & e_1^\top & -1 \\
& D_2 & & & & & e_2^\top & -1 \\
& & \ddots & & & & \vdots & \vdots \\
& & & D_N & & & e_N^\top & -1
\end{bmatrix}
\tag{12}
$$

where $F_i = I_{m_i} \otimes \mathbf{1}_m^\top$, $G_i^{(0)} = \mathbf{1}_{m_i}^\top \otimes I_m$, $D_i = \mathtt{vec}(D^{(i)})$, $e_i \in \mathbb{R}^N$ is the indicator vector of the $i$-th coordinate, for $i = 1, ..., N$.

Noticing that the last $N$ rows are linearly independent, and are linearly independent to the linear space spanned by other rows due to the last $N + 1$ rows, it suffices to prove that $A_0 := A(1 : n_c - N, 1 : n_v - N - 1)$, the matrix consisting of the non-zero columns of first $n_c - N$ rows in $A$, has full row rank. Now we only need to verify the following two fact: a) After a series of row transformations, we can transform matrix $A_0$ into a matrix $\bar{A}_0$ whose elements in $(M + 1)$-th, $(M + m + 1)$-th, $\cdots$, $(M + (N - 1)m + 1)$-th rows are zeros, and elements in other positions are the same as $A_0$. b) The matrix $\bar{A}_0$ has full row rank. The proof follows Lemma 3.1 of Ge et al. (2019), which we include here for completeness.

a). Define

$$
e_1 = \begin{bmatrix} 1 \\ 0 \\ \vdots \\ 0 \end{bmatrix}_{m \times 1}, \quad
T_i = \begin{bmatrix} 1 & 1 & \cdots & 1 \\ & & & \\ & & & \end{bmatrix}_{m \times m_i}, \quad
S_i = \begin{bmatrix} 1 & 1 & \cdots & 1 \\ & 1 & & \\ & & \ddots & \\ & & & 1 \end{bmatrix}_{m \times m}, \quad i = 1, ..., N
$$

and

$$
L_1 = \begin{bmatrix}
I_{m_1} & & & & & & \\
& \ddots & & & & & \\
& & I_{m_N} & & & & \\
-T_1 & & & I_m & & & \\
& \ddots & & & \ddots & & \\
& & -T_N & & & I_m & \\
& & & & & & 1
\end{bmatrix}, \quad
L_2 = \begin{bmatrix}
I_{m_1} & & & & & & \\
& \ddots & & & & & \\
& & I_{m_N} & & & & \\
& & & S_1 & & & e_1 \\
& & & & \ddots & & \vdots \\
& & & & & S_N & e_1 \\
& & & & & & 1
\end{bmatrix}
$$

Then

$$
L_2 L_1 A_0 =
\begin{bmatrix}
F_1 & & & & & & \\
& F_2 & & & & & \\
& & \ddots & & & & \\
& & & F_N & & & \\
G_1^{(1)} & & & & & & H^{(1)} \\
& G_2^{(1)} & & & & & H^{(1)} \\
& & \ddots & & & & \vdots \\
& & & G_N^{(1)} & & & H^{(1)} \\
& & & & & & \mathbf{1}_m^T
\end{bmatrix},
$$

where $G_i^{(1)} = S_i G_i^{(0)} - S_i T_i F_i$, $H^{(1)} = e_1 \mathbf{1}_m^\top - S_i$. Note that elements in the first rows of $H^{(1)}$ and $G_i^{(1)}$, $i = 1, ..., N$ are zeros. We have proved the claims in a).

b). $\bar{A}_0$ is obtained by removing the $(M+1)$-th, $(M+m+1)$-th, $\cdots$, $(M+(N-1)m+1)$-th rows of $A_0$. That is,

$$
\bar{A}_0 =
\begin{bmatrix}
F_1 & & & & & & \\
& F_2 & & & & & \\
& & \ddots & & & & \\
& & & F_N & & & \\
G_1^{(2)} & & & & & & H^{(2)} \\
& G_2^{(2)} & & & & & H^{(2)} \\
& & \ddots & & & & \vdots \\
& & & G_N^{(2)} & & & H^{(2)} \\
& & & & & & \mathbf{1}_m^\top
\end{bmatrix}
$$

where $G_i^{(2)} = G_i^{(1)}(2:m,:) = \mathbf{1}_{m_i}^\top \otimes [\mathbf{0}_{m-1}, I_{m-1}]$, $H^{(2)} = H^{(1)}(2:m,:) = [\mathbf{0}_{m-1}, -I_{m-1}]$ and $F_i = I_{m_i} \otimes \mathbf{1}_m^\top$. Let $n'_{row} = M + N(m-1) + 1$.

For $i = 1, ..., N$, let

$$
U_i = I_{m_i} \otimes
\begin{bmatrix}
1 & -1 & \cdots & -1 \\
& 1 & & \\
& & \ddots & \\
& & & 1
\end{bmatrix}_{m \times m},
\quad
U_{N+1} =
\begin{bmatrix}
1 & -1 & \cdots & -1 \\
& 1 & & \\
& & \ddots & \\
& & & 1
\end{bmatrix}_{m \times m},
\quad
R_1 =
\begin{bmatrix}
U_1 & & & \\
& U_2 & & \\
& & \ddots & \\
& & & U_{N+1}
\end{bmatrix}
$$

then

$$
\bar{A}_0 R_1 =
\begin{bmatrix}
F_1^{(3)} & & & & & & \\
& F_2^{(3)} & & & & & \\
& & \ddots & & & & \\
& & & F_N^{(3)} & & & \\
G_1^{(3)} & & & & & & H^{(3)} \\
& G_2^{(3)} & & & & & H^{(3)} \\
& & \ddots & & & & \vdots \\
& & & G_N^{(3)} & & & H^{(3)} \\
& & & & & & \boldsymbol{\alpha}^\top
\end{bmatrix}
$$

where $F_i^{(3)} = F_i U_i = I_{m_i} \otimes [1, \mathbf{0}_{m-1}^\top]$, $G_i^{(3)} = G_i^{(2)} U_i = G_i^{(2)} = \mathbf{1}_{m_i}^\top \otimes [\mathbf{0}_{m-1}, I_{m-1}]$, $i = 1, ..., N$, $H^{(3)} = H^{(2)} U_{N+1} = H^{(2)} = [\mathbf{0}_{m-1}, -I_{m-1}]$ and $\boldsymbol{\alpha}^\top = \mathbf{1}_m^\top U_{N+1} = [1, \mathbf{0}_{m-1}^\top]$.

Let

$$
\tilde{K} = \begin{bmatrix} 0 \\ & -1 \\ & & \ddots \\ & & & -1 \end{bmatrix}_{m \times m} , \quad K_i = \begin{bmatrix} I_m & \tilde{K} & \cdots & \tilde{K} \\ & I_m \\ & & \ddots \\ & & & I_m \end{bmatrix}_{mm_i \times mm_i} , \quad R_2 = \begin{bmatrix} K_1 \\ & \ddots \\ & & K_N \\ & & & I_m \end{bmatrix}
$$

then

$$
\bar{A}_0 R_1 R_2 = \begin{bmatrix} F_1^{(4)} \\ & F_2^{(4)} \\ & & \ddots \\ & & & F_N^{(4)} \\ G_1^{(4)} & & & & H^{(3)} \\ & G_2^{(4)} & & & H^{(3)} \\ & & \ddots & & \vdots \\ & & & G_N^{(4)} & H^{(3)} \\ & & & & \boldsymbol{\alpha}^\top \end{bmatrix}
$$

where $F_i^{(4)} = F_i^{(3)} K_i = F_i^{(3)} = I_{m_i} \otimes [1, \mathbf{0}_{m-1}^\top]$, $G_i^{(4)} = G_i^{(3)} K_i = [\mathbf{0}_{m-1}, I_{m-1}, \mathbf{0}_{(m-1) \times (mm_i - m)}]$, $i = 1, ..., N$,

Let $\tilde{A}_0$ be the matrix comsisting of the first $(mM + 1)$ columns of $\bar{A}_0 R_1 R_2$. That is,

$$
\tilde{A}_0 = \begin{bmatrix} F_1^{(4)} \\ & F_2^{(4)} \\ & & \ddots \\ & & & F_N^{(4)} \\ G_1^{(4)} \\ & G_2^{(4)} \\ & & \ddots \\ & & & G_N^{(4)} \\ & & & & 1 \end{bmatrix}
$$

The matrix $\tilde{A}_0$ possesses the following two properties:

1. Every row of $\tilde{A}_0$ contains exactly one nonzero entry equal to 1, while all remaining entries are zero;

2. Every column of $\tilde{A}_0$ contains no more than one nonzero entry.

Given these properties, we can establish the existence of permutation matrices $P_1 \in \mathbb{R}^{M+M'}$ and $Q_1 \in \mathbb{R}^{mM+1}$ such that the product $P_1 \tilde{A}_0 Q_1$ yields the form $[I_{(M+M')}, \mathbf{0}, \ldots, \mathbf{0}]$. Consequently, we have $\text{rank}(\tilde{A}_0) = \text{rank}(P_1 \tilde{A}_0 Q_1) = M + M'$, which implies $\text{rank}(\bar{A}_0) = M + M'$. This demonstrates that $\bar{A}_0$ achieves full row-rank.

## C. Algorithm for (WBC-LP): Predictor-Corrector Inner Point Method

---

**Algorithm 2** Predictor-Corrector Inner Point Method for Linear Programming (WBC-LP)

---

1: **Input**: The standard LP form of Problem (WBC-LP)

$$\min \quad c^T x$$
$$\text{s.t.} \quad \bar{A}x = b, \quad x \geq 0$$

2: **Initialization**: Set initial feasible point $(x_0, y_0, s_0)$, where $x_0 > 0$, $s_0 > 0$ (dual variables). Choose tolerance $\epsilon > 0$ and set iteration counter $k = 0$.

3: **while** $\|r_b\| > \epsilon$ and $\|r_c\| > \epsilon$ **do**

4:     Compute residuals:

$$r_b = \bar{A}x - b \quad \text{(primal residual)}$$
$$r_c = \bar{A}^T y + s - c \quad \text{(dual residual)}$$
$$r_s = Xs \quad \text{(complementarity residual)}$$

    where $X = \text{diag}(x)$, $S = \text{diag}(s)$, and $\mu = \frac{x^T s}{n}$ is the duality measure.

5:     **Predictor Step**: Solve the linear system for affine scaling direction $(\Delta x^{\text{aff}}, \Delta y^{\text{aff}}, \Delta s^{\text{aff}})$:

$$\begin{bmatrix} 0 & \bar{A}^T & I \\ \bar{A} & 0 & 0 \\ S & 0 & X \end{bmatrix} \begin{bmatrix} \Delta x^{\text{aff}} \\ \Delta y^{\text{aff}} \\ \Delta s^{\text{aff}} \end{bmatrix} = - \begin{bmatrix} r_c \\ r_b \\ r_s \end{bmatrix}$$

    We have

$$\Delta y^{\text{aff}} = -(\bar{A}R\bar{A}^\top)^{-1}(r_c - \bar{A}R(s - r_b)), \text{solved by calling Algorithm 1.} \tag{13}$$
$$\Delta s^{\text{aff}} = r_b - \bar{A}^\top \Delta y^{\text{aff}}, \quad \Delta x^{\text{aff}} = R(s - \Delta s).$$

6:     Compute the step size $\alpha_{\text{aff}}$ by finding the maximum step length that maintains $x + \alpha_{\text{aff}}\Delta x^{\text{aff}} \geq 0$ and $s + \alpha_{\text{aff}}\Delta s^{\text{aff}} \geq 0$.

7:     **Corrector Step**: Compute the corrector directions using central path perturbation with updated $r_s$:

$$r_s = r_s + \Delta X^{\text{aff}}\Delta s^{\text{aff}} - \mu \mathbf{1}$$

    and solve the system again to get $(\Delta x^{\text{corr}}, \Delta y^{\text{corr}}, \Delta s^{\text{corr}})$:

$$\Delta y^{\text{corr}} = -(\bar{A}R\bar{A}^\top)^{-1}\bar{A}(S^{-1}\Delta X^{\text{aff}}\Delta s^{\text{aff}} - \mu S^{-1}\mathbf{1}), \text{solved by calling Algorithm 1.} \tag{14}$$
$$\Delta s^{\text{corr}} = -\bar{A}^\top \Delta y^{\text{corr}}, \quad \Delta x^{\text{corr}} = S^{-1}\Delta X^{\text{aff}}\Delta s^{\text{aff}} - \mu S^{-1}\mathbf{1}.$$

8:     Compute the total search direction:

$$\Delta x = \Delta x^{\text{aff}} + \Delta x^{\text{corr}}, \quad \Delta y = \Delta y^{\text{corr}}, \quad \Delta s = \Delta s^{\text{aff}} + \Delta s^{\text{corr}}$$

9:     Compute the step size $\alpha$ by updating with both predictor and corrector directions.

10:     Update variables:

$$x_{k+1} = x_k + \alpha\Delta x, \quad y_{k+1} = y_k + \alpha\Delta y, \quad s_{k+1} = s_k + \alpha\Delta s.$$

11:     Update the duality measure $\mu$ and increment the iteration counter $k = k + 1$.

12: **end while**

13: **Output**: Optimal solution $(x^*, y^*, s^*)$.

---

# D. Proof of Proposition 3.4

*Proof.* Let $M := \sum_{i=1}^{N} m_i$ and $M' := N(m-1)$. Using the same notation as the preceding section,

$$\bar{A} = \begin{bmatrix} F_1 & & & & & & & & \\ & F_2 & & & & & & & \\ & & \ddots & & & & & & \\ & & & F_N & & & & & \\ G_1^{(2)} & & & & & H^{(2)} & & & \\ & G_2^{(2)} & & & & H^{(2)} & & & \\ & & \ddots & & & \vdots & & & \\ & & & G_N^{(2)} & H^{(2)} & & & & \\ & & & & \mathbf{1}_m^\top & & & & \\ D_1 & & & & & e_1^\top & -1 & & \\ & D_2 & & & & e_2^\top & & -1 & \\ & & \ddots & & & \vdots & & & \vdots \\ & & & D_N & & e_N^\top & & & -1 \end{bmatrix}$$

where $G_i^{(2)} = G_i^{(1)}(2:m,:) = \mathbf{1}_{m_i}^\top \otimes [\mathbf{0}_{m-1}, I_{m-1}]$, $H^{(2)} = H^{(1)}(2:m,:) = [\mathbf{0}_{m-1}, -I_{m-1}]$ and $F_i = I_{m_i} \otimes \mathbf{1}_m^\top$.

Let

$$\bar{G}_1 := \bar{A}(1:M,:) = \begin{bmatrix} F_1 & & & \\ & F_2 & & \\ & & \ddots & \\ & & & F_N \end{bmatrix},$$

$$\bar{G}_2 := \bar{A}(M+1:M+M',:) = \begin{bmatrix} G_1^{(2)} & & & & H^{(2)} \\ & G_2^{(2)} & & & H^{(2)} \\ & & \ddots & & \vdots \\ & & & G_N^{(2)} & H^{(2)} \end{bmatrix},$$

$$\bar{A}_3 := \bar{A}(M+M'+1,:) = \begin{bmatrix} \mathbf{0}, & \mathbf{0}, & \cdots, & \mathbf{0}, & \mathbf{1}_m^\top \end{bmatrix},$$

$$\bar{A}_4 = \bar{A}(M+M'+2:n_c-N,:) = \begin{bmatrix} D_1 & & & & e_1^\top & -1 \\ & D_2 & & & e_2^\top & -1 \\ & & \ddots & & \vdots & \vdots \\ & & & D_N & e_N^\top & -1 \end{bmatrix}.$$

Then

$$\bar{A} = \begin{bmatrix} \bar{G}_1 \\ \bar{G}_2 \\ \bar{A}_3 \\ \bar{A}_4 \end{bmatrix} \ and \ \bar{A}R\bar{A}^\top = \begin{bmatrix} \bar{G}_1 R \bar{G}_1^\top & \bar{G}_1 R \bar{G}_2^\top & \bar{G}_1 R \bar{A}_3^\top & \bar{G}_1 R \bar{A}_4^\top \\ \bar{G}_2 R \bar{G}_1^\top & \bar{G}_2 R \bar{G}_2^\top & \bar{G}_2 R \bar{A}_3^\top & \bar{G}_2 R \bar{A}_4^\top \\ \bar{A}_3 R \bar{G}_1^\top & \bar{A}_3 R \bar{G}_2^\top & \bar{A}_3 R \bar{A}_3^\top & \bar{A}_3 R \bar{A}_4^\top \\ \bar{A}_4 R \bar{G}_1^\top & \bar{A}_4 R \bar{G}_2^\top & \bar{A}_4 R \bar{A}_3^\top & \bar{A}_4 R \bar{A}_4^\top \end{bmatrix}.$$

Now we analyze the structure of each sub-matrix $\bar{A}_i R \bar{A}_j^\top$ and rename them for conciseness. Let $R = \mathtt{diag}(R_1, \cdots, R_{N+3})$, where $R_i \in \mathbb{R}^{mm_i \times mm_i}$, $i = 1, \ldots, N$, $R_{N+1} \in \mathbb{R}^{m \times m}$, $R_{N+2} \in \mathbb{R}^{N \times N}$, $R_{N+3} = r_{n_v}$. Then

$$\bar{G}_1 R \bar{G}_1^\top = \begin{bmatrix} F_1 R_1 F_1^\top & & \\ & \ddots & \\ & & F_N R_N F_N^\top \end{bmatrix} := B_1.$$

Each $F_i R_i F_i^\top$ is a diagonal matrix with positive diagonal entries.

$$\bar{G}_2 R \bar{G}_1^\top = \begin{bmatrix} G_1^{(2)} R_1 F_1^\top & & \\ & \ddots & \\ & & G_N^{(2)} R_N F_N^\top \end{bmatrix} := B_2^\top,$$

$$\bar{G}_2 R \bar{G}_2^\top = \begin{bmatrix} G_1^{(2)} R_1 G_1^{(2)\top} & & \\ & \ddots & \\ & & G_N^{(2)} R_N G_N^{(2)\top} \end{bmatrix} + \begin{bmatrix} H^{(2)} R_{N+1} H^{(2)\top} & \cdots & H^{(2)} R_{N+1} H^{(2)\top} \\ \vdots & & \vdots \\ H^{(2)} R_{N+1} H^{(2)\top} & \cdots & H^{(2)} R_{N+1} H^{(2)\top} \end{bmatrix}. \qquad (15)$$

where $H^{(2)} R_{N+1} H^{(2)\top}$ and each $G_i^{(2)} R_i G_i^{(2)\top}$ is a diagonal matrix with positive diagonal entries. We use $B_3$ to denote the first matrix in the right hand side of (15) and $B_4$ to denote the second. In addition, other blocks of $\bar{A} R \bar{A}^\top$ are

$$\bar{A}_3 R \bar{G}_1^\top = 0,$$

$$\bar{A}_3 R \bar{G}_2^\top = \begin{bmatrix} \mathbf{1}_m^\top R_{N+1} H^{(2)\top} & \cdots & \mathbf{1}_m^\top R_{N+1} H^{(2)\top} \end{bmatrix} := \boldsymbol{\alpha}^\top,$$

$$\bar{A}_3 R \bar{A}_3^\top = \mathbf{1}_m^\top R_{N+1} \mathbf{1}_m := l.$$

$$\bar{A}_4 R \bar{G}_1^\top = \begin{bmatrix} D_1 R_1 F_1^\top & & \\ & \ddots & \\ & & D_N^{(2)} R_N F_N^\top \end{bmatrix} := K_1^\top,$$

$$\bar{A}_4 R \bar{G}_1^\top = \begin{bmatrix} D_1^{(2)} R_1 F_1^\top & & \\ & \ddots & \\ & & D_N^{(2)} R_N F_N^\top \end{bmatrix} := K_2^\top,$$

$$\bar{A}_4 R \bar{A}_4^\top = \texttt{diag}(D_1 R_1 D_1^\top, D_2 R_2 D_2^\top, \cdots, D_N R_N D_N^\top) + R_{N+2} + r_{n_v} \mathbf{1}_N \mathbf{1}_n^\top := W,$$

here we have $W_1 := \texttt{diag}(D_1 R_1 D_1^\top, D_2 R_2 D_2^\top, \cdots, D_N R_N D_N^\top) + R_{N+2}$.

With all notations above, we have

$$\bar{A} R \bar{A}^T = \begin{bmatrix} B_1 & B_2 & \mathbf{0} & K_1 \\ B_2^\top & B_3 + B_4 & \boldsymbol{\alpha} & K_2 \\ \mathbf{0} & \boldsymbol{\alpha}^\top & l & 0 \\ K_1^\top & K_2^\top & 0 & W \end{bmatrix}$$

$\square$

# E. Derivation of Eq. (8)

$$\begin{bmatrix} I_M & & & \\ -B_2^\top B_1^{-1} & I_{M'} & & \\ & & 1 & \\ & & & I_N \end{bmatrix} \cdot \bar{A} R \bar{A}^T \cdot \begin{bmatrix} I_M & & & \\ -B_2^\top B_1^{-1} & I_{M'} & & \\ & & 1 & \\ & & & I_N \end{bmatrix}^\top = \begin{bmatrix} B_1 & & \mathbf{0} & K_1 \\ & G_1 + B_4 & \boldsymbol{\alpha} & \overline{K}_2 \\ \mathbf{0} & \boldsymbol{\alpha}^\top & l & 0 \\ K_1^\top & \overline{K}_2^\top & 0 & W \end{bmatrix}$$

$$\begin{bmatrix} I_M & & & \\ & I_{M'} & -\boldsymbol{\alpha}/l & \\ & & 1 & \\ & & & I_N \end{bmatrix} \begin{bmatrix} B_1 & & \mathbf{0} & K_1 \\ & G_1 + B_4 & \boldsymbol{\alpha} & \overline{K}_2 \\ \mathbf{0} & \boldsymbol{\alpha}^\top & l & 0 \\ K_1^\top & \overline{K}_2^\top & 0 & W \end{bmatrix} \begin{bmatrix} I_M & & & \\ & I_{M'} & -\boldsymbol{\alpha}/l & \\ & & 1 & \\ & & & I_N \end{bmatrix}^\top = \begin{bmatrix} B_1 & & \mathbf{0} & K_1 \\ & G_1 + G_2 & \mathbf{0} & \overline{K}_2 \\ \mathbf{0} & \mathbf{0}^\top & l & 0 \\ K_1^\top & \overline{K}_2^\top & 0 & W \end{bmatrix}$$

$$\begin{bmatrix} I_M & & & \\ & I_{M'} & & \\ & & 1 & \\ -K_1^\top B_1^{-1} & & & I_N \end{bmatrix} \begin{bmatrix} B_1 & & \mathbf{0} & K_1 \\ & G_1 + G_2 & \mathbf{0} & \overline{K}_2 \\ \mathbf{0} & \mathbf{0}^\top & l & 0 \\ K_1^\top & \overline{K}_2^\top & 0 & W \end{bmatrix} \begin{bmatrix} I_M & & & \\ & I_{M'} & & \\ & & 1 & \\ -K_1^\top B_1^{-1} & & & I_N \end{bmatrix}^\top = \begin{bmatrix} B_1 & & & \\ & G_1 + G_2 & & \overline{K}_2 \\ & & l & \\ & \overline{K}_2^\top & & W \end{bmatrix}.$$

## F. Proof for Theorem 3.5

The proof follows the same approach of Lemma 4.2 in Ge et al. (2019), which is to prove positive definite property by confirming strict diagonal dominance. This approach requires several foundational results about the underlying matrix structure.

**Lemma F.1.** *The block matrices $B_1, B_2, B_3, B_4$ possess the following fundamental properties:*

*(i) All entries are non-negative.*

*(ii) The identity $B_3 \mathbf{1}_{M_2} = B_2^\top \mathbf{1}_M$ holds.*

*(iii) The strict inequality $B_1 \mathbf{1}_M - B_2 \mathbf{1}_{M_2} > \mathbf{0}$ is satisfied.*

*Proof.* Property (i) follows directly from the construction using non-negative matrices $\bar{A}$ and $R$.

For property (ii), we compute:

$$B_3 \mathbf{1}_{M_2} = \begin{bmatrix} G_1^{(2)} R_1 G_1^{(2)\top} \mathbf{1}_{m-1} \\ \vdots \\ G_N^{(2)} R_N G_N^{(2)\top} \mathbf{1}_{m-1} \end{bmatrix}$$

and

$$B_2^\top \mathbf{1}_M = \begin{bmatrix} G_1^{(2)} R_1 F_1^\top \mathbf{1}_{m_1} \\ \vdots \\ G_N^{(2)} R_N F_N^\top \mathbf{1}_{m_N} \end{bmatrix}$$

Using the definitions $G_i^{(2)} = \mathbf{1}_{m_i}^\top \otimes [\mathbf{0}_{m-1}, I_{m-1}]$ and $F_i = I_{m_i} \otimes \mathbf{1}_m^\top$, along with the diagonal structure of $R_i$, we establish that $G_i^{(2)} R_i F_i^\top \mathbf{1}_{m_i} = G_i^{(2)} R_i G_i^{(2)\top} \mathbf{1}_{m-1}$, yielding the desired identity.

For property (iii), observe that:

$$B_1 \mathbf{1}_M = \begin{bmatrix} F_1 R_1 F_1^\top \mathbf{1}_{m_1} \\ \vdots \\ F_N R_N F_N^\top \mathbf{1}_{m_N} \end{bmatrix}, \quad B_2 \mathbf{1}_{M_2} = \begin{bmatrix} F_1 R_1 G_1^{(2)\top} \mathbf{1}_{m-1} \\ \vdots \\ F_N R_N G_N^{(2)\top} \mathbf{1}_{m-1} \end{bmatrix}$$

Since $F_i R_i F_i^\top \mathbf{1}_{m_i} > F_i R_i G_i^{(2)\top} \mathbf{1}_{m-1}$ for each block $i$, the component-wise strict inequality follows. $\square$

Now we proceed to establish Theorem 3.5.

***Proof of Theorem 3.5.*** The block-diagonal structure of $G_1$ is immediate from its construction. We establish positive definiteness via contradiction.

Assume $G_1$ is not positive definite. Then there exists an eigenvalue $-\lambda \leq 0$, making the matrix $\lambda I_{M_2} + G_1$ singular.

Consider the vector equation:

$$(\lambda I_{M_2} + G_1) \mathbf{1}_{M_2} = \lambda \mathbf{1}_{M_2} + B_3 \mathbf{1}_{M_2} - (B_2^\top B_1^{-1} B_2) \mathbf{1}_{M_2} \tag{16}$$

Applying property (ii) from Lemma F.1:

$$= \lambda \mathbf{1}_{M_2} + B_2^\top \mathbf{1}_M - (B_2^\top B_1^{-1} B_2) \mathbf{1}_{M_2}$$
$$= \lambda \mathbf{1}_{M_2} + B_2^\top B_1^{-1} B_1 \mathbf{1}_M - (B_2^\top B_1^{-1} B_2) \mathbf{1}_{M_2}$$
$$= \lambda \mathbf{1}_{M_2} + B_2^\top B_1^{-1} (B_1 \mathbf{1}_M - B_2 \mathbf{1}_{M_2}) \tag{17}$$

Since $B_2^\top B_1^{-1} \geq 0$ with each row containing at least one strictly positive element, and property (iii) ensures $B_1 \mathbf{1}_M - B_2 \mathbf{1}_{M_2} > \mathbf{0}$, we conclude:

$$(\lambda I_{M_2} + G_1) \mathbf{1}_{M_2} > \mathbf{0}_{M_2}$$

Next, we analyze the structure of $\lambda I_{M_2} + G_1 = \lambda I_{M_2} + B_3 - B_2^\top B_1^{-1} B_2$. Since $B_1, B_2, B_3 \geq 0$ and $B_3$ is diagonal, this matrix has positive diagonal entries and non-positive off-diagonal entries.

Define the projection operators:

$$D_{M_2} := I_{M_2} \quad \text{(diagonal mask)} \tag{18}$$

$$O_{M_2} := \mathbf{1}_{M_2} \mathbf{1}_{M_2}^\top - I_{M_2} \quad \text{(off-diagonal mask)} \tag{19}$$

Then:

$$D_{M_2} \circ |\lambda I_{M_2} + G_1| = D_{M_2} \circ (\lambda I_{M_2} + G_1) \tag{20}$$

$$O_{M_2} \circ |\lambda I_{M_2} + G_1| = -O_{M_2} \circ (\lambda I_{M_2} + G_1) \tag{21}$$

where $\circ$ denotes the Hadamard product. This yields:

$$[D_{M_2} \circ |\lambda I_{M_2} + G_1| - O_{M_2} \circ |\lambda I_{M_2} + G_1|] \mathbf{1}_{M_2} = (\lambda I_{M_2} + G_1) \mathbf{1}_{M_2} > \mathbf{0}_{M_2}$$

This establishes strict diagonal dominance of $\lambda I_{M_2} + G_1$, implying non-singularity and contradicting our assumption. Therefore, $G_1$ is positive definite.

As for $G_2$, direct computation shows that:

$$G_2 = (\mathbf{1}_N \mathbf{1}_N^\top) \otimes \left( \text{diag}(\boldsymbol{z}) - \frac{1}{l} \boldsymbol{z} \boldsymbol{z}^\top \right)$$

Given that $l > \mathbf{1}_{m-1}^\top \boldsymbol{z}$ and $\boldsymbol{z} \geq \mathbf{0}$ by construction, set $Y = \text{diag}(\boldsymbol{z}) - \frac{1}{l} \boldsymbol{z} \boldsymbol{z}^\top$, we know that $Y$ is positive definite by similar proof as that of $G_1$. $\qquad \square$

## G. Algorithm Complexity

***Proof of Theorem 3.2.*** We analyze the time and storage complexities of Algorithm 1 separately.

**Time complexity.** We list the most time consuming part of each step corresponding to Algorithm 1.

In step 1, computing of $B_1, B_3, W, Q$ takes most flops of $O(\sum_{i=1}^N m^2 m_i)$, achieved when computing $B_1$ and $B_3$.

In step 2, computing of $(G_1 + G_2)^{-1}$ takes most flops of $O(Nm^3)$, due to the need to invert $G_{ii}, i = 1, \cdots, N$; computing of $\tilde{W}$ takes $O(N^2 m^2)$ flops, which may exceed other procedures on time complexity when $N$ is sufficiently large.

In step 3, computing $\tilde{W}^{-1}$ takes $O(N^3)$ flops, while other part takes most $O(Nm^2)$.

To sum up, the total time complexity is $O(m^2 \sum_{i=1}^N m_i + Nm^3 + N^2 m^2 + N^3)$.

**Space complexity.** For the implementation of the whole interior-point methods, the major data that should be kept in the memory include the following four parts:

**(a)** Compute primal variables or dual variables. Note that the size of a primal variable is $O(m(\sum_{i=1}^N m_i))$, and a dual variable is of size $O(M + M')$.

**(b)** Matrix $\bar{A}$. According to the structure of $\bar{A}$ shown in appendix D, the matrix contains several sparse blocks, and we only need to store the non-zero entries of each block. Overall, the space for storing $\bar{A}$ is bounded by $2m \sum_{i=1}^N m_i + N(m-1) + m$. Since each column of $E_1$ and $E_2$ has at most one non-zero term, the total number of non-zero entries in $E_1$ and $E_2$ is bounded by $2m \sum_{i=1}^N m_i$. In addition, $E_3$ has $m - 1$ non-zero terms, $D$ has $m \sum_{i=1}^N m_i$ non-zero terms as the i-th row of it contains $m m_i$ non-zero terms. So the total number of non-zero terms in $\bar{A}$ is bounded by $O(M)$.

**(c)** Diagonals of matrices $B_1$ and $B_3$, and diagonal blocks of matrices $K_1, K_2, B_2$ and $G_1$. $B_1$ and $B_3$ takes $n_c - N = O(M + M')$ doubles. The diagonal blocks of matrices $B_2$ and $G_1$ have $m \sum_{i=1}^N m_i$ elements and $N(m-1)^2$ elements, respectively. Noticing that every row of $D'$ only has one non-zero term, we have $K_1$ of size smaller than $nnz(E_1) = O(M)$, the number of nonzero terms of $E_1$, and $K_2$ smaller than $nnz(E_2) = O(M)$.

**(d)** The matrix $W, \bar{W}$ and $\tilde{W}$. Takes no more than $N^2$ to store.

**(e)** Other intermediate vectors or matrices, whose data scale is bounded by a constant time of the data scale in (a), (b), (c) and (d).

With the analysis in (a) to (d), we know the memory usage of algorithm 1 is bounded by $O(m \sum_{i=1}^N m_i + Nm^2 + N^2)$.

$\square$

## H. Woodbury identity

Given the matrices $A, T, C$, and $V$, the standard formula for Woodbury identity (Hager, 1989) is

$$(A + TCV)^{-1} = A^{-1} - A^{-1}T(C^{-1} + VA^{-1}T)^{-1}VA^{-1}.$$

In the proof of Lemma 3.6, the corresponding $A, T, C$, and $V$ are $G_1, \mathbf{1}_N \otimes U^\top, I_{M'}, \mathbf{1}_N^\top \otimes U$, respectively.

## I. The free support WBC

For the free support WB, many previous researches (Cuturi & Doucet, 2014; Ge et al., 2019; Huang et al., 2021) apply block coordinate descent, which requires minimizing the

$$
\begin{aligned}
\min_{\boldsymbol{w}, X, \{\Pi^{(t)}\}} \quad & \sum_{t=1}^N \left\langle D(X, Q^{(t)}), \Pi^{(t)} \right\rangle \\
\text{s.t.} \quad & \Pi^{(t)} \mathbf{1}_{m_t} = \boldsymbol{w}, \left(\Pi^{(t)}\right)^\top \mathbf{1}_m = \boldsymbol{a}^{(t)}, \Pi^{(t)} \geq 0, \forall t = 1, \cdots, N \\
& \mathbf{1}_m^\top \boldsymbol{w} = 1, \boldsymbol{w} \geq 0
\end{aligned}
\tag{22}
$$

where $\boldsymbol{w} := (w_1, \cdots, w_m)^\top \in \mathbb{R}_+^m, X := [\boldsymbol{x}_1, \cdots, \boldsymbol{x}_m] \in \mathbb{R}^{d \times m_t}, \Pi^{(t)} \in \mathbb{R}_+^{m \times m_t}$ and $D(X, Q^{(t)}) := [\|\boldsymbol{x}_i - \boldsymbol{q}_j^{(t)}\|^2] \in \mathbb{R}^{m \times m_t}$ for $t = 1, \cdots, N$. Problem (22) is a nonconvex problem, where one needs to find the optimal support points X and the optimal weight vector $\boldsymbol{w}$ of a barycenter simultaneously. But i specified from the support points of $\{\mathcal{P}^{(t)}\}_{t=1}^N$.

For the free support Wasserstein barycenter, many previous researches (Cuturi & Doucet, 2014; Ge et al., 2019; Huang et al., 2021) apply block coordinate descent. For our Wasserstein ball center, the objective becomes:

$$
\begin{aligned}
\min_{\boldsymbol{w}, X, \{\Pi^{(t)}\}} \max_{t \in [N]} \quad & \left\langle D(X, Q^{(t)}), \Pi^{(t)} \right\rangle \\
\text{s.t.} \quad & \Pi^{(t)} \mathbf{1}_{m_t} = \boldsymbol{w}, \left(\Pi^{(t)}\right)^\top \mathbf{1}_m = \boldsymbol{a}^{(t)}, \Pi^{(t)} \geq 0, \forall t = 1, \cdots, N \\
& \mathbf{1}_m^\top \boldsymbol{w} = 1, \boldsymbol{w} \geq 0
\end{aligned}
\tag{23}
$$

where $\boldsymbol{w} := (w_1, \cdots, w_m)^\top \in \mathbb{R}_+^m, X := [\boldsymbol{x}_1, \cdots, \boldsymbol{x}_m] \in \mathbb{R}^{d \times m_t}, \Pi^{(t)} \in \mathbb{R}_+^{m \times m_t}$ and $D(X, Q^{(t)}) := [\|\boldsymbol{x}_i - \boldsymbol{q}_j^{(t)}\|^2] \in \mathbb{R}^{m \times m_t}$ for $t = 1, \cdots, N$. Problem (22) is a nonconvex problem. By block coordinate descent, one optimise the support set $X$, and then the fixed support WBC to obtain the weight $\boldsymbol{w}$ of WBC and coupling matrices $\Pi^{(t)}$ alternately. The algorithm will converges to a local minima. For instance, in $l_2$ space, the minimization of $X$ is $\min_X \max_{t \in [N]} \sum_{k=1}^m \sum_{j=1}^{m_t} \|\boldsymbol{x}_k - \boldsymbol{q}_j^{(t)}\|^2 \pi_{kj}^{(t)}$.

This is a quadratically constrained quadratic program (QCQP) in that it can be reformulated as the following problem:

$$
\begin{aligned}
\min_{X, \zeta} \quad & \zeta \\
\text{s.t.} \quad & \sum_{k=1}^m \sum_{j=1}^{m_t} \|\boldsymbol{x}_k - \boldsymbol{q}_j^{(t)}\|^2 \pi_{kj}^{(t)} \leq \zeta
\end{aligned}
\tag{24}
$$

Since the quadratic forms of $X$ in the constraints is positive semidefinite, the problem is convex [1], thus can be efficiently solved with convex programming, such as interior point method. Note that the size of variable $X, \zeta$ is $m + 1$, the number of constraints is $N$, the scale of solving $X$ is much smaller than solving the fixed support WBC.

# J. Supplementary Experiment

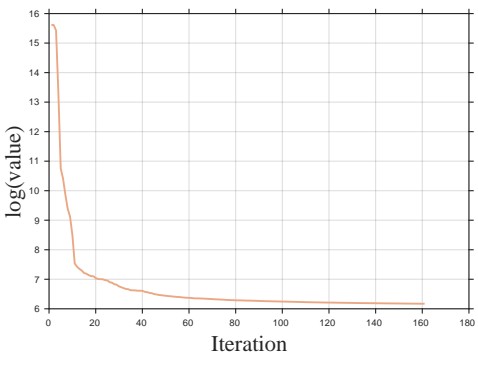

*Figure 5.* N=10, m=6000

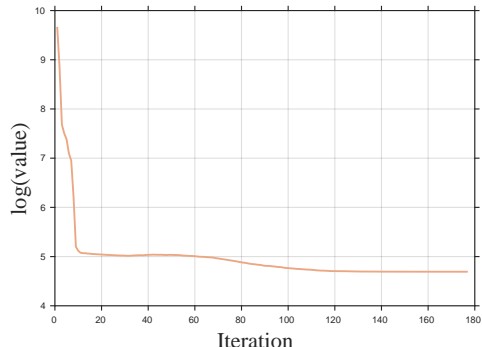

*Figure 6.* N=2000, m=300

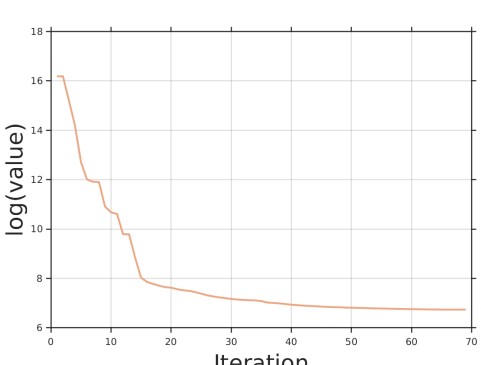

*Figure 7.* N=3, m=30000

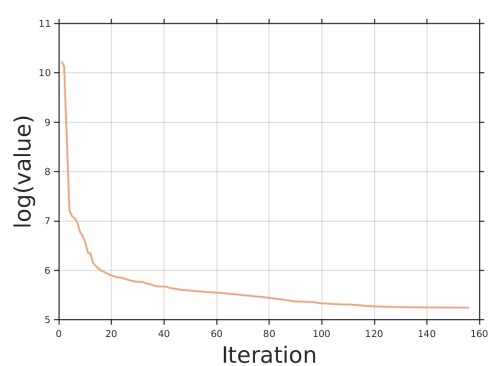

*Figure 8.* N=500, m=500

## J.1. Super-linearity of our algorithm

A sequence $\{x_n\}$ is said to converge *super-linearly* to a limit $x^*$ if:

$$\lim_{n \to \infty} \frac{|x_{n+1} - x^*|}{|x_n - x^*|} = 0$$

This means that the error $|x_n - x^*|$ decreases at a rate faster than linear convergence. With our logarithmic coordinate of objective value, we have shown that our algorithm converges in super-linearly speed,

In both case when $m \gg N$ and $m \ll N$, the super-linear convergence speed maintains.

## J.2. Fairness of WBC

To compare the imf, we set all distributions to be on support $[m] \subset \mathbb{R}^1$. For distributions in family 1, we assign the points in $\{1, 2, \cdots, 10\}$ with probability proportional to independent samples from $U([3, 4])$, the uniform distribution on $[3, 4]$, and other points with probability proportional to samples from $U([0, 1])$. For a distribution in family 2, we assign the points in $\{m - 9, \cdots, m\}$ with probability proportional to independent samples $U([3, 4])$, and other points with probability proportional to samples from $U([0, 1])$. See Fig. 9 for the instance tested in Fig. 3(b): We also conduct several other experiments with different settings, see Figs. 10 and 11.

To evaluate and compare the var(WD), we consider a comprehensive set of experimental configurations. The experiments are conducted across the following settings:

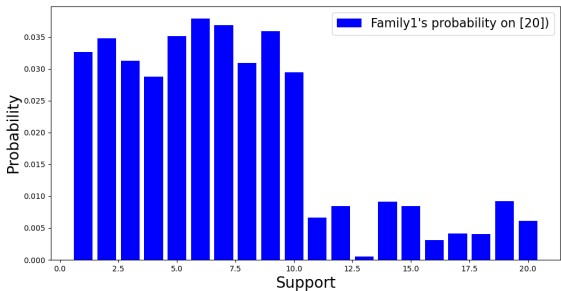 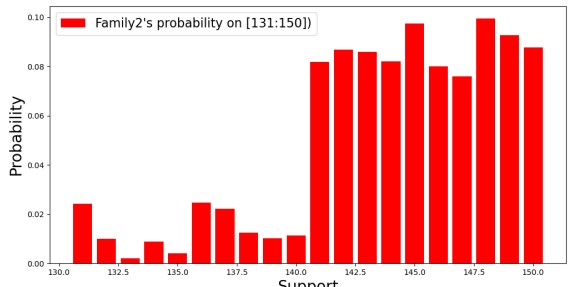

*Figure 9.* The left histogram shows a probability in family 1 on [20], whose density is concentrated in [10]. The right is a probability in family 2, concentrated in $[m-9, m]$.

- Ground space dimensions: 3, 10, and 100.

- Ground metrics: $\ell_1$, $\ell_2$, and $\ell_\infty$.

- Support selection regimes:

  - Uniform sampling within a cube;
  - Gaussian sampling;

  Each sampling regime is further augmented with cluster-diversifying transformations, which divide the support points into 1, 2, or 3 clusters.

For every combination of the above three parameter categories, we generate 50 independent instances, resulting in a total of 2700 experimental runs. Every instance consists of 30 probability distributions, each of which has weights sampled uniformly and support size of 200.

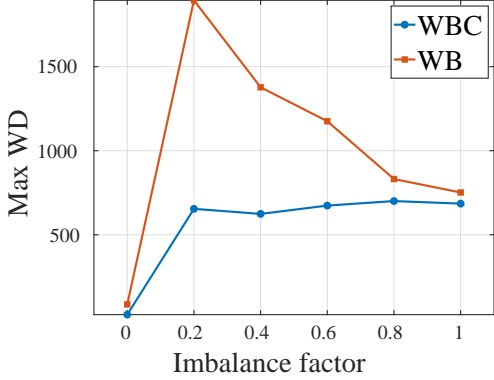 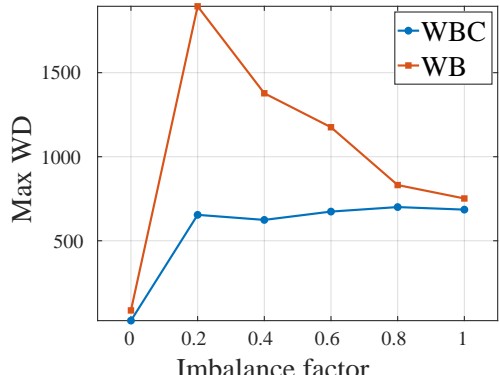

*Figure 10.* N=40, m=150                    *Figure 11.* N=15, m=600

### J.3. The Point cloud data

This is a free-support problem. We select the support with a two-step strategy: firstly, choose the support from a combination of a dense grid and samples from a normal distribution with large variance, computing the weight of WBC; then update the support of WBC by adding the support in last step with samples near the points who have particular large weights, computing the final WBC. Both WB and WBC are given grey scales proportional to the weight of the point, so more weight means paler color.

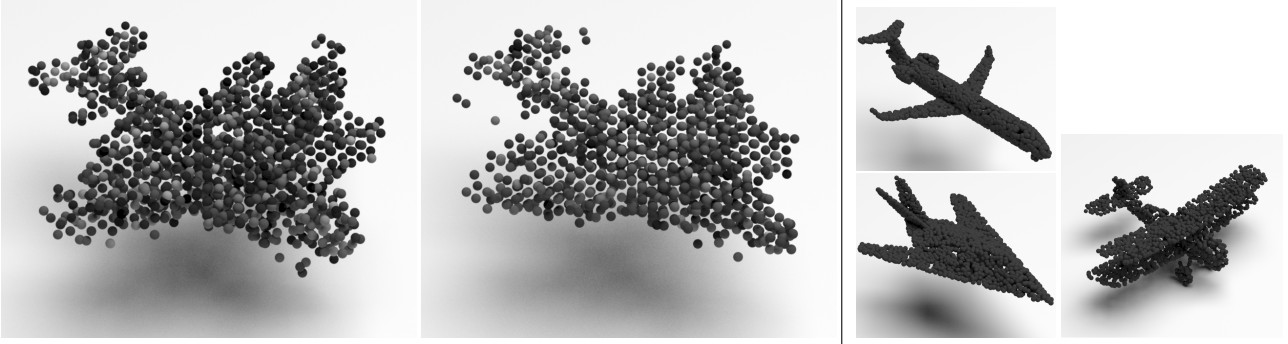

*Figure 12.* WBC (left) and WB (middle) of 3 airplane, one of which is a biplane (with a double-decker wing structure), while the other two are both monoplanes. Compared to the WB, the WBC retains more characteristics of the biplane.

