# OpenReview forum: "Finding Wasserstein Ball Center: Efficient Algorithm and The Applications in Fairness"
_ICML.cc/2025/Conference — ICML 2025 poster_

### Official Review · Reviewer_YCD8 · 2025-03-06

**Overall Recommendation:** 4

**Summary:**

This paper considers fairness in representing a set of distributions and proposes to use the Wasserstein Ball Centers (WBC) as a representative of a distribution instead of the Wasserstein Barycenter (WB). Given a set of distributions $\mu_1, \ldots, \mu_N$, the Wasserstein Ball Center is defined as a distribution $\mu$ that minimizes the maximum distance to all distributions $\mu_1, \ldots, \mu_N$. Although the definition of WBC seems to be new, as the authors mentioned, the concept of Wasserstein balls is not new.

The main contribution of this paper is in designing an efficient algorithm for computing the WBC of a set of discrete distributions, given a support for the WBC distribution. In particular, they present an IPM-based algorithm that computes the WBC in $O(Nm^3 + N^2m^2 + N^3) time, where $N$ is the number of distributions and $m$ is the support size of each distribution.


## Update after rebuttal
The authors resolved my concerns by adding additional experimental results comparing the disparity scores of their proposed barycenter distribution and the well-known Wasserstein barycenter. I increased my score from 3 to 4.

**Claims And Evidence:**

The Theorems and Lemmas are clear.

**Essential References Not Discussed:**

No

**Experimental Designs Or Analyses:**

The experimental results show some benefits of the proposed method over the Gurobi solver. I have some questions about the experiments on the running times (experiment 1):
How did you form the distributions? What are the dimensions of the distributions? Are they all supported on the same support? How did you choose the support of the WBC?

**Methods And Evaluation Criteria:**

I have concerns with the fairness evaluation criteria, as the only criterion used for measuring the fairness of the proposed method is the maximum distance of each group to the representative distribution (which is exactly the objective function that the WBC tries to minimize). It is not surprising to have lower maximum Wasserstein distance of each group to WBC compared to WB, and although one of the metrics for computing fairness is the maximum distance, many other metrics need be used to show better fairness of the proposed method over WB.

I understand that the authors claim the main contribution of their paper to be the design of fast algorithms for computing WBC; however, since the authors provide experiments on comparing the fairness of their method compared to WB, they would need to provide more evidence on that front to consider it an added value to the paper.

**Other Comments Or Suggestions:**

No

**Other Strengths And Weaknesses:**

The main weakness of the presented work is that the paper assumes that the support of the WBC is already given. It is not clear at all how one can derive this support set, and it is not the same as the support of the distributions. For instance, suppose we have three distributions in 2D, each one supported on a single point, forming a triangle. Then, one needs to take points inside this triangle while forming the WBC.

**Questions For Authors:**

I asked my questions in previous sections.

**Relation To Broader Scientific Literature:**

The Wasserstein barycenter is a widely used method for representing a set of distributions and has found applications in numerous scientific fields. However, it might not be the fairest representation as it might be far from some under-represented groups. This paper suggested the use of WBC instead to resolve the fairness and provided an algorithm for computing WBC.

**Theoretical Claims:**

I checked some of the proofs, and they seem to be correct. I have a problem with the construction of b, and I would appreciate it if the authors would let me know whether I am missing something.

In the construction of the matrix A, the first M + Nm rows check the mass preservation of the transport plans. Then there is a 1_m, which is supposed to check that the total mass of w is 1. The next N rows ensure that gamma_i's are properly defined; *here, I think there is a missing 1_N in the vector b*.

---

> ### Author Rebuttal · Authors · 2025-04-01
>
> **Q1** On the missing part of the vector $\boldsymbol{b}$.
>
>  We sincerely appreciate the reviewer’s meticulous reading. Yes, there is a missing $\boldsymbol{0}_N$ as the last $N$ terms of vector $\boldsymbol{b}$. We will correct this typo in the final version. Notebly, since the design of our algorithm focuses on the constraint matrix $A$, this typo does not affect any other part of the paper.
>
> **Q2** About the experiments on the running times (experiment 1): How did you form the distributions? What are the dimensions of the distributions? Are they all supported on the same support? How did you choose the support of the WBC?
>
>   For the first three question, see Line 403-405 in paper: "we generate random datasets in $\mathbb{R}$, and the weights of $(q_1^{(t)}, ..., q_m^{(t)})$ in each distribution $P^{(t)}$ are generated uniformly at random."
>
> The support points of the WBC is generated in the same way, and we will clarify this in the final draft. Thank you for pointing out this ambiguity.
>
> **Q3** About the fairness metric in paper
>
>  We sincerely appreciate your valuable suggestion, which highlights the need for a more comprehensive discussion of fairness in the context of WBC.
>
> One of the inspirations of this paper is the *social fairness* concept as mentioned in the last paragraph of Introduction and Appendix A. Apart from that, *Minimax fairness* is gaining increasing attention within the machine learning community [1-3], where fairness is achieved by minimizing the maximum error across groups—as opposed to minimizing the average error (i.e., replacing $\min\sum$ with $\min\max$ in the objective). This aligns precisely with our formulation.
>
> We acknowledge that there are other fairness notions—such as demographic parity [4], equalized odds/opportunity [5-6], and individual fairness [7]. Extending these frameworks to probability-measure spaces (e.g., via Wasserstein metrics) presents a compelling direction for future research, and we will include a brief discussion of this in the revised manuscript.
>
> **Q4** How can one derive this support set, and it is not the same as the support of the distributions?
>
> Due to space limitations, please refer to our response to **Q2** of Reviewer f8bo to view block coordinate descent method. As for our point-cloud experiment (see Appendix I.3), we proceeds as follows:
>
> 1. **Initialization**: Compute the fixed-support WBC using the vertices of a dense grid as the initial support set.
> 2. **Support Update**:
>    - *Augmentation*: Sample new points from Gaussian distributions centered at high-weight support points.
>    - *Pruning*: Remove support points with negligible weights (below a threshold).
> 3. **Iteration**: Recompute the fixed-support WBC on the updated support and repeat until convergence.
>
> This design is motivated by two key observations:
>
> - The WB admits a sparse support representation, with a theoretical upper bound of $N\max\limits_{i\in [N]}m_i$ [8].
> - Adaptive refinement balances computational efficiency with solution accuracy by dynamically focusing on high-density regions, which avoid too many iterations.
>
> From the theoretical perspective, "free-support WBC'' is indeed a much more challenging problem than the fixed-support case, especially when the dimensionality is high. We further note that even the ``fixed-support WB'' problem is also very hard and the polynomial-time algorithm with high precision is only for fixed-dimension [8]. We appreciate the reviewer for raising this question, and will spend more effort to investigate this problem as the future work.
>
> [1] Martinez, Natalia, Martin Bertran, and Guillermo Sapiro. "Minimax pareto fairness: A multi objective perspective." *International conference on machine learning*. PMLR, 2020.
>
> [2] Abernethy, Jacob D., et al. "Active Sampling for Min-Max Fairness." *International Conference on Machine Learning*. PMLR, 2022.
>
> [3] Singh, Harvineet, et al. "When do minimax-fair learning and empirical risk minimization coincide?." *International Conference on Machine Learning*. PMLR, 2023.
>
> [4] Zemel, Rich, et al. "Learning fair representations." *International conference on machine learning*. PMLR, 2013.
>
> [5] Hardt, Moritz, Eric Price, and Nati Srebro. "Equality of opportunity in supervised learning." *Advances in neural information processing systems* 29 (2016).
>
> [6]  Woodworth, B., Gunasekar, S., Ohannessian, M., and Srebro, N. (2017). Learning non-discriminatory predictors. In Proceedings of the 2017 Conference on Learning Theory, pages 1920–1953.
>
> [7] Dwork, Cynthia, et al. "Fairness through awareness." *Proceedings of the 3rd innovations in theoretical computer science conference*. 2012.
>
> [8] Altschuler, Jason M., and Enric Boix-Adsera. "Wasserstein barycenters can be computed in polynomial time in fixed dimension." *Journal of Machine Learning Research* 22.44 (2021): 1-19.

---

> > ### Comment · Reviewer_YCD8 · 2025-04-02
> >
> > I thank the authors for their thorough response.
> >
> > Concerning the fairness metric, there exist some other fairness metrics, such as the Disparity Impact as the variance or range of the distances of the distributions to the barycenter and the Normalized Disparity Score.
> >
> > I also believe that to compare the fairness, the experiments can be enhanced a lot by considering more meaningful distributions in higher distributions rather than 1D uniform distribution.

---

> > > ### Author Response · Authors · 2025-04-05
> > >
> > > **Q1** About other fairness metric.
> > >
> > > Thank you for your insightful suggestion regarding additional disparity impact analysis, and we will incorporate further discussion and experiments on this topic to the final manuscript. Here, we conducted some preliminary experiments to compare the variance of the Wasserstein distances from the input distributions to the WBC (denoted as $\textrm{var}(d_{WBC})$) and that of the WB (denoted as $\textrm{var}(d_{WB})$). Below are the settings in our experiment:
> > >
> > > - Ground space dimensions: 3, 10 and 100.
> > > - Ground metrics: $L_1, L_2$, and $L_\infty$.
> > > - Support selection regimes: Uniform sampling in a cube, Gaussian sampling. Both composed with cluster-diversifying transformations (which divide the support points into 1, 2 or 3 clusters).
> > >
> > > For each parameter combination of above three categories, we test 50 different instances (2700 total instances). Each instance contains 30 distributions with uniform sampled weights, and each distribution has support size 200. In *all​* 2700 instances, $\textrm{var}(d\_{WBC}) < \textrm{var}(d\_{WB})$, indicating that **$\text{var}(d\_{WBC})$ is usually smaller than $\textrm{var} (d\_{WB})$.** For example,  in $L\_1$ space of dimension 100, we sample the 30 supports uniformly in a cube with side length 3, then translate 3 supports by adding $-\boldsymbol{1}\_{100}$, and translate 5 supports by adding $2*\boldsymbol{1}\_{100}$, run algorithms to obtain that $\textrm{var}(d\_{WBC})=58.35$, while $\textrm{var}(d\_{WB})=418.61$. Those observation suggests that WBC not only improves fairness for the minority distributions, but also could enhance fairness *across all distributions* by limiting disparities between transport plans. To further explain this phenomenon, we will explore the theory of the inherent property of transport plans to WBC in future work.
> > >
> > > **Q2** Test other distributions in higher dimensions to compare fairness.
> > >
> > > Thanks for your suggestion. In our paper, the fairface experiment (Section 4(3)) and the pointcloud experiment (Section 4(4)) showcases the fairness performance of WBC in real-world data distributions in $\mathbb{R}^2$ and $\mathbb{R}^3$. Here, we further conduct a set of additional experiments in higher dimension that generalise the "fairness of WBC" experiment in Section 4(2). The experimental distributions are partitioned into two families, where the supports of family 1 are sampled from normal distributions centered at $5*\boldsymbol{1}\_d$ and those in family 2 are sampled from normal distributions centered at $-5*\boldsymbol{1}\_d$, $d$ is the dimension of ground space. To quantify the disparity between these two families, define the imbalanced factor ("imf'') as the ratio of the first family's size to the second one. Results are listed in table 1-3. The results demonstrate fairness on those distributions in higher dimension, and we will include more experiments of high dimension in the final manuscript.
> > >
> > > Table 1: fairness in $L_2$ space of dimension 100. We notice that the magnitude defference between $\textrm{var}(d_{WBC})$ and $\textrm{var}(d_{WB})$ is big. This phenomenon may be related to properties of normal distributions in high-demonsional space, and will be studied later.
> > > | imf           | 0     | 0.2    | 0.4    | 0.6    | 0.8    | 1     |
> > > | ------------- | ----- | ------ | ------ | ------ | ------ | ----- |
> > > | Max WD of WBC | 48.24 | 281.25 | 301.41 | 228.64 | 228.42 | 194.52 |
> > > | Max WD of WB  | 61.54 | 578.70 | 548.55 | 447.46 | 292.54 | 218.15 |
> > > |$\textrm{var}(d_{WBC})/10^{-9}$| 8.76| 56.50|63.22|53.28| 64.89|54.81|
> > > |$\textrm{var}(d_{WB})/10^3$|8.54| 7.51| 26.41|36.21|9.20|5.67|
> > >
> > > Table 2: fairness in $L_2$ space of dimension 1000.
> > > | imf           | 0     | 0.2    | 0.4    | 0.6    | 0.8    | 1     |
> > > | ------------- | ----- | ------ | ------ | ------ | ------ | ----- |
> > > | Max WD of WBC$/10^3$ | 1.44 | 1.48 | 1.47 | 1.46 | 1.51 | 1.45 |
> > > | Max WD of WB$/10^3$| 1.85 | 2.75 | 2.11 | 1.87 | 1.84 | 1.79|
> > > |$\textrm{var}(d_{WBC})/10^{-10}$| 6.84| 14.50|7.26|11.04| 5.18|4.41|
> > > |$\textrm{var}(d_{WB})/10^4$|5.13| 27.55| 32.48|41.50|9.21|16.61|
> > >
> > > Table 3: fairness in $L_2$ space of dimension 10000.
> > > | imf           | 0     | 0.2    | 0.4    | 0.6    | 0.8    | 1     |
> > > | ------------- | ----- | ------ | ------ | ------ | ------ | ----- |
> > > | Max WD of WBC$/10^4$ | 1.25 | 1.35 | 1.41 | 1.37 | 1.32 | 1.30 |
> > > | Max WD of WB$/10^4$ | 1.54 | 1.90 | 1.75 | 1.60 | 1.43 | 1.48 |
> > > |$\textrm{var}(d_{WBC})/10^{-10}$| 8.76| 5.54|6.06|13.26| 4.88|4.81|
> > > |$\textrm{var}(d_{WB})/10^4$|3.54| 7.19| 31.24|46.20|40.25|11.38|
> > >
> > > We also supplement the runtime analysis in Section 4(1) with high-dimensional experiments, which will be discussed in detail in the final manuscript.
> > >
> > > Table 4: When $N=30, d=1000$, running time (s) as the support size $m$ varies.
> > > |m /100| 1|2|3|4|5|6|7|
> > > |---|--|---|--|--|--|--|--|
> > > |Ours|4.56 |5.42| 11.07| 12.49|  26.15 |30.83|35.81|
> > > |Gurobi| 17.11|53.21|410.02| 726.84| 1650.14|Null | Null|

---

### Official Review · Reviewer_f8bo · 2025-03-09

**Overall Recommendation:** 3

**Summary:**

This paper introduces the concept of "Wasserstein Ball Center" (WBC) as an alternative to the traditional Wasserstein Barycenter (WB) for finding a representative probability distribution from multiple input distributions. While WB minimizes the sum of Wasserstein distances from the barycenter to all input distributions, WBC minimizes the maximum Wasserstein distance to any input distribution. This "minmax" approach makes WBC more fair to "minority" distributions that differ significantly from the majority.
The main contributions are:

Formulating the WBC problem as finding the center of the minimum-radius ball that covers all input distributions in Wasserstein space

Showing that the fixed-support WBC problem can be formulated as a large-scale linear programming (LP) instance

Developing an efficient interior point method (IPM) to solve this LP problem by exploiting structure in the constraint matrix to achieve significant speed improvements over standard approaches

Demonstrating through experiments that WBC provides more equitable treatment of outlier distributions compared to WB

The authors' algorithm accelerates the IPM by O(min{N²m, Nm², m⁴}) times compared to vanilla IPM, where N is the number of distributions and m is the support size. Experiments on both synthetic and real datasets show that their algorithm is more computationally efficient than commercial solvers and that WBC indeed provides more fair representation for minority distributions.

**Claims And Evidence:**

The paper's claims are generally well-supported by evidence:

1. The theoretical claims about computational complexity improvements are supported by detailed mathematical derivations and proofs that exploit the structure of the linear programming formulation.

2. The fairness advantage of WBC over WB is demonstrated through:

- Visual examples showing how WBC preserves characteristics of outlier distributions (Fig. 1)
- Quantitative results showing reduced maximum Wasserstein distance for minority distributions (Fig. 3b)
- Case studies using the FairFace dataset showing more equitable treatment across racial groups
- 3D point cloud visualization showing how WBC preserves features from minority shapes


The computational efficiency claims are validated by comparing runtime performance against the Gurobi commercial solver, with clear advantages demonstrated across various problem scales.

**Essential References Not Discussed:**

The paper is generally thorough in its literature review, but a few relevant references might strengthen the connections:

Recent advances in fast approximation algorithms for Wasserstein distance computation. For exmaple (https://arxiv.org/abs/2312.01432)

Some recent work on fair Wasserstein barycenters (e.g., weighted schemes that give more importance to minority groups) could provide interesting contrast to the proposed minmax approach.

**Experimental Designs Or Analyses:**

The experimental designs and analyses in the paper are sound:

Compare the proposed algorithm against Gurobi, a state-of-the-art commercial solver

Evaluate performance with varying N (number of distributions) and m (support size)

Measure both computation time and objective value

Demonstrate super-linear convergence rate (Fig. 3a)

**Methods And Evaluation Criteria:**

The proposed methods and evaluation criteria are appropriate for the problem at hand:

1. The paper provides a sound mathematical formulation of the WBC problem as a linear program, which is a natural approach for solving the proposed minmax optimization.

2. The interior point method (IPM) with specialized matrix manipulations is an appropriate choice for this problem, as it exploits the specific structure of the constraint matrix.

3. The evaluation criteria are well-chosen: Computational efficiency is measured against a strong baseline (Gurobi solver)

**Other Comments Or Suggestions:**

NA

**Other Strengths And Weaknesses:**

Strengths:

The paper addresses an important problem - the inherent bias in Wasserstein barycenters against minority distributions - with a principled mathematical approach.

The experimental results convincingly demonstrate both computational efficiency and fairness benefits across differents scenarios.

The paper connects theoretical optimal transport concepts with practical fairness implications, especially in domains like medical imaging where fairness is crucial.


Weaknesses:

The paper focuses primarily on the discrete, fixed-support case. Some discussion on how the approach might extend to continuous distributions or free-support problems would strengthen the work.

The paper could provide more guidance on choosing between WB and WBC for different applications - when is the fairness benefit of WBC worth potential increases in average Wasserstein distance?

The computational improvements, while significant, are still limited by the inherent complexity of the LP formulation. For very large-scale problems, approximation methods might be necessary.  (https://arxiv.org/abs/2312.01432)

The experimental evaluation could benefit from more downstream tasks to show how the improved representation of minority distributions impacts practical applications.

**Questions For Authors:**

How does the proposed WBC approach handle noisy distributions or outliers that might not represent meaningful minorities but rather data corruption?

**Relation To Broader Scientific Literature:**

The paper builds upon and extends several areas in the scientific literature:

Wasserstein Barycenter

Fairness in Machine Learning

Optimal Transport

**Theoretical Claims:**

The paper contains several theoretical claims and proofs, particularly around the computational complexity of their algorithm and the structure of the constraint matrix in the proposed LP formulation.

The main theoretical result is Theorem 3.2, which states that the time complexity of each inner iteration of the IPM is O(m²∑ᵢmᵢ + Nm³ + N²m² + N³), and the memory usage is O(m∑ᵢmᵢ + Nm² + N²).

The proof of this theorem is built on several intermediate results:

Proposition 3.1 (showing the constraint matrix can be made full row-rank)

Proposition 3.4 (describing the structure of the normal equation matrix)

Proposition 3.5 (further decomposing the structure of specific submatrices)

Lemma 3.6 (providing an efficient way to compute a key matrix inverse)

---

> ### Author Rebuttal · Authors · 2025-04-01
>
> **Q1:** Missing references
>
>  Thank you for your valuable suggestions on the references, and we will add them to the revised version.
>
>
> **Q2:** Algorithm for free-support Wasserstein ball center (WBC)
>
>  For the free support Wasserstein barycenter, many previous researches [1-3] apply block coordinate descent. For our WBC, we can also utilize this approach, where the objective becomes:
> \\[
> \begin{array}{cl}
> & \min\limits_{\boldsymbol{w}, X, \left\\{ \Pi^{(t)} \right\\}} \max\limits_{t\in [N]} \left\langle D(X,Q^{(t)}), \Pi^{(t)} \right\rangle \\\\
> \text{s.t.} & \Pi^{(t)} \boldsymbol{1}_{m_t} = \boldsymbol{w}, \left( \Pi^{(t)} \right)^{\top} \boldsymbol{1}_m = \boldsymbol{a}^{(t)}, \Pi^{(t)} \geq 0, \forall t = 1, \cdots, N \\\\
> & \boldsymbol{1}_m^{\top} \boldsymbol{w} = 1, \boldsymbol{w} \geq 0
> \end{array}
> \\]
>
> where $\boldsymbol{w} := (w_1, \cdots, w_m)^{\top} \in \mathbb{R}_+^m$, $X:= [\boldsymbol{x}\_1, \cdots, \boldsymbol{x}\_m] \in \mathbb{R}^{d \times m\_t}$, $\Pi^{(t)} \in \mathbb{R}\_+^{m\times m\_t}$ and $D(X, Q^{(t)}):= [ \\| \boldsymbol{x}_i - \boldsymbol{q}^{(t)}_j \\|^p ]\in \mathbb{R}^{m \times m_t}$ for $t=1,\cdots, N$.
>
> Free support WBC is nonconvex. By block coordinate descent, one optimizes the support set $X$, and then the fixed support WBC to obtain the weight $\boldsymbol{w}$ of WBC and coupling matrices $\Pi^{(t)}$ alternately. The algorithm will converge to a local minima. For instance, in $l_2$ space, the minimization of $X$ is $\min\limits\_{X}\max\limits\_{t\in [N]}\sum\_{k=1}^m\sum\_{j=1}^{m\_t}||\boldsymbol{x}\_k-\boldsymbol{q}\_j^{(t)}||^2\pi\_{kj}^{(t)}$. This is a quadratically constrained quadratic program (QCQP) in that it can be reformulated as the following problem:
>
> \\[
> \begin{aligned}
> &\min\limits\_{X,\zeta} \zeta \\\\
> \text{s.t.} \ \ &\sum_{k=1}^m\sum\_{j=1}^{m_t}||\boldsymbol{x}\_k-\boldsymbol{q}\_j^{(t)}||^2\pi\_{kj}^{(t)}\leq \zeta
> \end{aligned}
> \\]
>
> Since the quadratic forms of $X$ in the constraints is positive semidefinite, the problem is convex [4], thus can be efficiently solved with convex programming, such as interior point method. Note that the size of variable $X, \zeta$ is $m+1$, the number of constraints is $N$, the scale of solving $X$ is much smaller than solving the fixed support WBC.
>
> Due to space limitations, please refer to our response to **Q4** of Reviewer YCD8 for our approach in the point-cloud experiment (see also Appendix I.3).
>
> From the theoretical perspective, "free-support WBC''  is indeed a much more challenging problem than the fixed-support case, especially when the dimensionality is high. We further note that even the ``fixed-support WB'' problem is also very hard and the polynomial-time algorithm with high precision is only for fixed-dimension [5-6]. We appreciate the reviewer for raising this question, and will spend more effort to investigate this problem as the future work.
>
>
> **Q3:** Provide more guidance on choosing between WB and WBC for different applications - when is the fairness benefit of WBC worth potential increases in average Wasserstein distance?
>
>  Thanks for this question, and we will include a discussion on it. In general, the trade-off between **fairness** (WBC) and **average performance** (WB) depends on the application:
>
> - **WBC** is preferable when minimizing *worst-case deviation* is critical (e.g., equitable resource allocation).
> - **WB** is better for applications where *average accuracy* dominates (e.g., density estimation).
>
> **Q4:** How does the proposed WBC approach handle noisy distributions or outliers that might not represent meaningful minorities but rather data corruption?
>
>  Thank you for raising this interesting question. Please see the answer to **Q2** in response to Reviewer Xncv.
>
> [1] Cuturi, Marco, and Arnaud Doucet. "Fast computation of Wasserstein barycenters." *International conference on machine learning*. PMLR, 2014.
>
> [2] Ge, Dongdong, et al. "Interior-point methods strike back: Solving the wasserstein barycenter problem." *Advances in neural information processing systems* 32 (2019).
>
> [3] Huang, Minhui, Shiqian Ma, and Lifeng Lai. "Projection robust Wasserstein barycenters." *International Conference on Machine Learning*. PMLR, 2021.
>
> [4] Floudas, Christodoulos A., and Viswanathan Visweswaran. "Quadratic optimization." *Handbook of global optimization* (1995): 217-269.
>
> [5] Altschuler, Jason M., and Enric Boix-Adsera. "Wasserstein barycenters can be computed in polynomial time in fixed dimension." *Journal of Machine Learning Research* 22.44 (2021): 1-19.
>
> [6] Lin, Tianyi, et al. "Fixed-support Wasserstein barycenters: Computational hardness and fast algorithm." Advances in neural information processing systems 33 (2020): 5368-5380.

---

### Official Review · Reviewer_Xncv · 2025-03-14

**Overall Recommendation:** 4

**Summary:**

This paper considers the following problem: Given a set of N probability measures, find a probability distribution that, minimizes the maximum distance to any input distribution. Intuitively, we can think of this as the problem of finding the center and radius of the “smallest Wasserstein ball” that encloses all the input distributions. A closely related problem is the Wasserstein barycenter problem where the objective is to minimize the sum total of the Wasserstein distances from the center to the $N$ distributions. While Wasserstein barycenter has been extensively studied, there is (to my knowledge) nothing known about solving the smallest Wasserstein ball problem.

Similar to the Wasserstein barycenter, if we assume that the support for the center is fixed, then one can formulate the exact problem as an LP and use standard methods to solve the problem. The authors make some interesting observations that gives this LP a simpler structure resulting in a faster exact algorithm using IPM for the problem. The overall execution time of their algorithm is $O(Nm^3+ N^2m^2+N^3)$ as opposed to $O(N^3m^4)$ using vanilla IPM.

The authors apply this novel optimization problem as a means to achieve fair centers to represent the distributions including those that may be underrepresented. The authors implement their solutions and show that they outperform state-of-the-art LP solvers (such as gurobi). They also provide some evidence on the impact of their approach on the problem of fairness.

Rebuttal Update: Thank you for your response. I will keep my positive score.

**Claims And Evidence:**

All claims are supported by proofs.

**Essential References Not Discussed:**

I did not find any missing references.

**Experimental Designs Or Analyses:**

Yes. The experimental set up seems sound to me.

**Methods And Evaluation Criteria:**

The authors show the benefits of proposed algorithm (in terms of time and memory usage) with a gurobi implementation. This is a solid comparison. The experiment to highlight fairness seems like a reasonable one.

**Other Comments Or Suggestions:**

NA

**Other Strengths And Weaknesses:**

Overall, the optimization problem considered here is novel and a very natural question to study. The result provided is interesting both from a theoretical standpoint as well as from an empirical standpoint.

My main concern that the paper does not address is the sensitivity of the objective function to the presence of an outlier distribution. A single outlier distribution can disproportionately shift the center towards the outlier. Due to this, I worry that the actual practical value of the optimization problem in real-world may be limited. Could the authors comment on this?

**Questions For Authors:**

Please address the concern raised in strength/weakness section.

**Relation To Broader Scientific Literature:**

The problem studied is novel and I'm not aware of any paper that directly addresses the algorithmic question of computing the smallest Wasserstein ball for fixed support. However, the problem is similar to the classical 1-center problem (perhaps the discrete 1-center problem because of the fixed support requirement) in Wasserstein metric. Since 1-center for arbitrary metric spaces is extensively studied, perhaps adding a comparison of your work with that would be good.

**Theoretical Claims:**

I have skimmed through the proofs presented in the main text and I did not find any issues with them.

---

> ### Author Rebuttal · Authors · 2025-04-01
>
> **Q1:** Comparison between our work and 1-center problem for arbitrary metric space.
>
> Thanks for the question regarding the connection between our WBC problem and the 1-center problem in arbitrary metric spaces. We will add more references and explanations in our paper. In Euclidean space, 1-center problem is often called “minimum enclosing ball problem'' or ”minmax location problem''. Their numerical or approximate algorithms has been extensively studied, see [1-5].  However, the 1-center problem under Wasserstein metric remains unexplored—precisely the gap our work aims to address. Due to the inherent nature and complexity of  Wasserstein metric, those previous 1-center algorithms cannot be directly applied to handle our proposed WBC problem (to our best knowledge).
>
>
> **Q2:** How to deal with outlier distributions?
>
>  Thank you for raising this interesting question. Actually, we think it is hard to precisely distinguish between "outlier'' and "minority'', if there is no any prior knowledge/assumption. For example,  a minority distribution could be far away from other distributions, and thus performs quite similarly as an outlier. So we believe an efficient way should take some prior knowledge on inliers and outliers into account.
>
> When no prior information is available, we can also utilize the following  iterative strategy to extract "outlier'' distributions and construct the WBC for the remaining distributions (which is similar with the "trim'' idea in statistics): initially, we compute the WBC of the input distributions; perform the following two steps in each iteration, until stable:
>
> 1. Recognize the set of distributions farthest from the current WBC as outliers;
> 2. Update the WBC as the WBC of the remaining distributions.
>
> Nevertheless, since we do not have any prior knowledge, the above strategy could bring some error (e.g., mistaken a minority distribution as an outlier). So we believe it deserves more further study on how to distinguish between "outlier'' and "minority'', and which kinds of prior knowledge can be utilized.
>
> [1] Bâdoiu, Mihai, and Kenneth L. Clarkson. "Smaller core-sets for balls." Proceedings of the fourteenth annual ACM-SIAM symposium on Discrete algorithms. 2003.
>
> [2] Tansel, B.Ç. (2011). Discrete Center Problems. In: Eiselt, H., Marianov, V. (eds) Foundations of Location Analysis. International Series in Operations Research & Management Science, vol 155. Springer, New York.
>
> [3] Abboud, Amir, et al. "On Complexity of 1-Center in Various Metrics." *Approximation, Randomization, and Combinatorial Optimization. Algorithms and Techniques* (2023).
>
> [4] Yildirim, E. Alper. "Two algorithms for the minimum enclosing ball problem." *SIAM Journal on Optimization* 19.3 (2008): 1368-1391.
>
> [5] Kumar, Piyush, Joseph SB Mitchell, and E. Alper Yildirim. "Approximate minimum enclosing balls in high dimensions using core-sets." *Journal of Experimental Algorithmics (JEA)* 8 (2003): 1-1.

---

### Decision · Program_Chairs · 2025-05-01

**Decision:**

Accept (poster)

**Comment:**

This paper presents the Wasserstein Ball Center  as a new method for identifying a representative probability distribution from multiple inputs. The proposed approach focuses on minimizing the maximum distance promoting fairness towards minority distributions compared to
Wasserstein barycenter which minimizes weighted average distance to distributions.
All reviewers agree that the contributions are original and solid and are happy to accept the paper.